# Aligning Text-to-Image Diffusion Models to Human Preference by Classification

**Longquan Dai, Xiaolu Wei, He Wang, Shaomeng Wang, and Jinhui Tang***
**Nanjing University of Science and Technology, Nanjing, China**
{dailongquan, weixiaolu, wanghe, smw, tangjinhui}@njust.edu.cn

## Abstract

Text-to-image diffusion models are typically trained on large-scale web data, often resulting in outputs that misalign with human preferences. Inspired by preference learning in large language models, we propose ABC (Alignment by Classification), a simple yet effective framework for aligning diffusion models with human preferences. In contrast to prior DPO-based methods that depend on suboptimal supervised fine-tuned (SFT) reference models, ABC assumes access to an ideal reference model perfectly aligned with human intent and reformulates alignment as a classification problem. Under this classification view, we recognize that preference data naturally forms a semi-supervised classification setting. To address this, we propose a data augmentation strategy that transforms preference comparisons into fully supervised training signals. We then introduce a classification-based ABC loss to guide alignment. Our alignment by classification approach could effectively steer the diffusion model toward the behavior of the ideal reference. Experiments on various diffusion models show that our ABC consistently outperforms existing baselines, offering a scalable and robust solution for preference-based text-to-image fine-tuning. Code is available at https://github.com/dailongquan/abc.

## 1 Introduction

Text-to-image diffusion models [4] have dominated image generation for years, trained on web-scale text-image pairs in a single stage. However, this approach may produce images misaligned with human preferences. In contrast, Large Language Models (LLMs) excel at generating human-preferred outputs through a two-stage process: pre-training on web data and fine-tuning on preference data. Applying this fine-tuning strategy to text-to-image models could enhance their ability to meet diverse user preferences, making them more useful and relevant.

Recent research [2, 9, 14, 59, 61] has focused on enhancing diffusion models to align with human preferences using Reinforcement Learning from Human Feedback (RLHF) [22]. This RLHF approach involves pretraining a reward model [56] to capture human preferences and then optimizing the diffusion models to maximize the reward of generated images. However, creating a robust reward model that accurately reflects human preferences is both challenging and computationally costly, and over-optimizing the reward model can lead to significant issues of model collapse [38].

Diffusion-DPO [51] integrates Direct Preference Optimization (DPO) [40] into the preference learning framework of diffusion models, eliminating the need for a reward model. DPO reparameterizes the reward function in RLHF to directly learn a model from preference data. In DPO, the implicit reward is formulated using the log ratio of the likelihood of a response between the current model and the supervised fine-tuned (SFT) model. However, the SFT model is far from an ideal model that aligns with human preferences completely. We hypothesize that this discrepancy may lead to suboptimal performance.

---

*Corresponding author.

39th Conference on Neural Information Processing Systems (NeurIPS 2025).

In this work, we propose the ABC (Alignment by Classification) framework, a simple yet effective preference optimization algorithm to solve this problem. The core of our algorithm relies on three key insights: (1), DPO with an ideal reference model can be framed as a classification problem using a diffusion model. (2), The alignment performance depends on the discriminative ability of the diffusion model. (3), We identify alignment with preference data as semi-supervised learning and propose a data augmentation method to convert it into supervised data. (4), We propose a classification-based ABC loss, incorporating augmented preference data, to align the diffusion model, which is equivalent to aligning the diffusion model with the ideal reference model. Therefore a reference model is not needed during training which saves a lot of memory.

We conduct empirical evaluations of our ABC framework on state-of-the-art text-to-image diffusion models, including SD1.5 [42] and SDXL) [37], comparing it with leading image preference alignment methods. Extensive analysis demonstrates that our ABC method effectively leverages preference data, resulting in a more accurate ranking of winning and losing responses.

## 2 Related Work

**Diffusion model alignment** can be achieved through fine-tuning [11, 60, 63]. Recently, methods [1, 2, 9, 14, 27, 48, 59, 61] based on RLHF have garnered increasing attention. Among them, Wallace et al. [51] expanded direct preference optimization [40], which was originally suggested for language models, to diffusion models, aligning them with pairwise preference datasets on top of a frozen reference model. Similarly, Li et al. [26] applies Kahneman-Tversky Optimization [13] from language model alignment to diffusion models to inject preferences into the reference model. Theoretically, using an ideal alignment model as the reference should produce the best results, but all these methods rely on a non-perfect SFT checkpoint as the reference model. In this paper, we disclose that using an ideal alignment model as the reference and minimizing the DPO loss will minimize a classification loss. We thus transform the reference-required alignment task into a reference-free classification task.

**Diffusion model classification** is a type of generative classification [64], where class probabilities $p(y|\boldsymbol{x})$ are inferred by modeling the data likelihood $p(\boldsymbol{x}|y)$ using generative models. Compared to discriminative classifiers [49], generative classifiers tend to be more robust and better calibrated [31]. Zimmermann et al. [65] leverage score-based models to compute the log-likelihood $p(\boldsymbol{x}|y)$ via integration and then apply Bayes' theorem to obtain $p(y|\boldsymbol{x})$. Other works [16, 21] perform diffusion in logit space to model the categorical classification distribution. Recent studies [6, 8, 17, 25] convert diffusion models into generative classifiers, showing that generative networks can be effectively repurposed for discriminative tasks. In this paper, we further reveal a close connection in the reverse direction: discriminative learning, specifically classification, can also be naturally applied to a particular generative task—diffusion model alignment.

**Classification loss** generally follows two paradigms: learning with class-level labels and learning with pairwise labels. In the first setting, the model is trained to assign each input to its corresponding class using a classification loss, such as L2-Softmax [41], Large-margin Softmax [29], Angular Softmax [28], NormFace [52], AM-Softmax [53], and ArcFace [12]. In contrast, pairwise-based approaches learn to directly model similarity or dissimilarity between sample pairs. Representative methods include contrastive loss [7, 15], triplet loss [19, 43], Lifted-Structure loss [34], N-pair loss [46], Histogram loss [50], Angular loss [54], Margin-based loss [57], and Multi-Similarity loss [55]. In this paper, we employ Circle loss [47], which unifies the two paradigms, to conduct the diffusion model alignment task.

## 3 Background

Diffusion models are certifiably robust classifiers [5]. To provide a classification perspective on the preference alignment of diffusion models, we offer a preliminary discussion on diffusion models [4, 18] and diffusion classifiers [5, 8], as well as the Circle loss [47], a generalized classification loss.

### 3.1 Diffusion Models

We briefly review denoising diffusion probabilistic models [18]. Given $\boldsymbol{x}_0$ from a real data distribution $q(\boldsymbol{x}_0)$ and assuming that the signal-to-noise ratio $\text{SNR}(t) = \alpha_t/\sigma_t^2$ is monotonically decreasing

over time, the forward diffusion process gradually adds Gaussian noise to the data to obtain a sequence of noisy samples $\{\boldsymbol{x}_t\}_{t=1}^T$ according to $\{\alpha_t\}_{t=1}^T$ and $\{\sigma_t\}_{t=1}^T$ which are designed such that $\boldsymbol{x}_T$ is nearly an Gaussian distribution $q(\boldsymbol{x}_t|\boldsymbol{x}_0) = \mathcal{N}(\boldsymbol{x}_t; \sqrt{\alpha_t}\boldsymbol{x}_0, \sigma_t^2 \mathbf{I})$. The reverse process $p_\theta(\boldsymbol{x}_{t-1}|\boldsymbol{x}_t) = \mathcal{N}(\boldsymbol{x}_{t-1}; \boldsymbol{\mu}_\theta(\boldsymbol{x}_t, t), \tilde{\sigma}_t^2 \mathbf{I})$ is defined as a Markov chain aimed at approximating $q(\boldsymbol{x}_0)$ by gradually denoising from the Gaussian distribution $p(\boldsymbol{x}_T) = \mathcal{N}(\boldsymbol{x}_T; \mathbf{0}, \boldsymbol{I})$. where $\boldsymbol{\mu_\theta}$ is generally parameterized by a time-conditioned noise prediction network $\boldsymbol{\epsilon_\theta}(\boldsymbol{x}_t, t)$. Let $C$ be a small constant and $w_t$ be the weight. The reverse process can be learned by optimizing the variational lower bound on the log-likelihood as

$$\log p_{\boldsymbol{\theta}}(\boldsymbol{x}) \geq -\mathbb{E}_{\epsilon, t}\left[ w_t \|\boldsymbol{\epsilon_\theta}(\boldsymbol{x}_t, \mathsf{y}, t) - \boldsymbol{\epsilon}\|_2^2 \right] + C \tag{1}$$

### 3.2  Classification with Diffusion Models

Consider a dataset $\Omega = \left\{ (\boldsymbol{x}^{(l)}, y^{(l)}) \right\}_{l=1}^L$, where each image $\boldsymbol{x}^{(l)}$ is associated with a label $y^{(l)}$ that belongs to one of $K$ classes, denoted as $\mathsf{Y} = \{\mathsf{y}_k\}_{k=1}^K$. Given a new image $\boldsymbol{x}$, the classification objective is to predict the class label $\tilde{y}$ that has the highest probability of being assigned to $\boldsymbol{x}$.

$$\tilde{y} = \mathrm{argmin}_{\mathsf{y}_k \in \mathsf{Y}} -p(\mathsf{y}|\boldsymbol{x}) = \mathrm{argmin}_{\mathsf{y} \in \mathsf{Y}} -p(\boldsymbol{x}|\mathsf{y}) \cdot p(\mathsf{y}). \tag{2}$$

Assuming a uniform prior distribution over the classes, i.e. $p(\mathsf{y}_k) = \frac{1}{K}$ for all $k$, the prior term becomes constant and can be ignored in the maximization process. Thus, the problem reduces to:

$$\tilde{y} = \mathrm{argmin}_{\mathsf{y} \in \mathsf{Y}} -\log p(\boldsymbol{x}|\mathsf{y}). \tag{3}$$

Clark and Jaini [8] leverage the score function $\bar{s}_{\boldsymbol{\theta}}(\boldsymbol{x}, \mathsf{y})$ (5), which can be considered a good measure of the similarity between the category prompt $\mathsf{y}$ and the image $\boldsymbol{x}$, to approximate $\log p_\theta(\boldsymbol{x}|\mathsf{y})$ and convert the text-to-image diffusion model into a classifier (4).

$$\tilde{y} = \mathrm{argmin}_{\mathsf{y} \in \mathsf{Y}} -\log p_\theta(\boldsymbol{x}|\mathsf{y}) \approx \mathrm{argmin}_{\mathsf{y} \in \mathsf{Y}} -\bar{s}_{\boldsymbol{\theta}}(\boldsymbol{x}, \mathsf{y}), \text{ where} \tag{4}$$

$$\bar{s}_{\boldsymbol{\theta}}(\boldsymbol{x}, \mathsf{y}) = -\mathbb{E}_{\epsilon, t}\left[ s_{\boldsymbol{\theta}}(\boldsymbol{x}, \mathsf{y}) \right] \text{ and } s_{\boldsymbol{\theta}}(\boldsymbol{x}, \mathsf{y}) = w_t \|\boldsymbol{\epsilon} - \boldsymbol{\epsilon_\theta}\left(\sqrt{\alpha_t}\boldsymbol{x} + \sigma_t \boldsymbol{\epsilon}, \mathsf{y}, t\right)\|_2^2 \tag{5}$$

Further, Chen et al. [5] extend this approach by calculating the class probability $p_\theta(\mathsf{y}|\boldsymbol{x})$ through

$$p_{\boldsymbol{\theta}}(\mathsf{y}_k|\boldsymbol{x}) = \frac{p(\boldsymbol{x}|\mathsf{y}_k) \cdot p(\mathsf{y}_k)}{\sum_{\mathsf{y}_j \in \mathsf{Y}} p(\boldsymbol{x}|\mathsf{y}_j) \cdot p(\mathsf{y}_j)} = \frac{\exp(\log p_\theta(\boldsymbol{x}|\mathsf{y}_k))}{\sum_{\mathsf{y}_j \in \mathsf{Y}} \exp(\log p_\theta(\boldsymbol{x}|\mathsf{y}_j))} \approx \frac{\exp(\bar{s}_{\boldsymbol{\theta}}(\boldsymbol{x}, \mathsf{y}_k))}{\sum_{\mathsf{y}_j \in \mathsf{Y}} \exp(\bar{s}_{\boldsymbol{\theta}}(\boldsymbol{x}, \mathsf{y}_j))} \tag{6}$$

### 3.3  Circle Loss for Classification

Classification involves selecting one target category from $K$ candidate categories. Suppose the scores of $\boldsymbol{x}$ for binary categories are $\{\mathsf{y}_i\}_{i=1}^2$ for simplicity. The binary cross-entropy loss is given by:

$$\mathcal{L}_{\mathrm{BCE}} = \sum_{\boldsymbol{x}} \left[ \log\left(1 + \exp\left(\iota(\boldsymbol{x}, \mathsf{y}_1, \mathsf{y}_2)\right)\right) + \log\left(1 + \exp\left(\iota(\boldsymbol{x}, \mathsf{y}_2, \mathsf{y}_1)\right)\right) \right], \text{ where}$$
$$\iota(\boldsymbol{x}, \mathsf{y}^+, \mathsf{y}^-) = \mathbb{E}_{\epsilon, t}\left[ s_{\boldsymbol{\theta}}(\boldsymbol{x}, \mathsf{y}^+) \right] - \mathbb{E}_{\epsilon, t}\left[ s_{\boldsymbol{\theta}}(\boldsymbol{x}, \mathsf{y}^-) \right]. \tag{7}$$

Sun et al. [47] propose the binary Circle loss $\mathcal{L}_{\mathrm{Circle}}$ by extending $\iota(\boldsymbol{x}, \mathsf{y}^+, \mathsf{y}^-)$ as:

$$\iota(\boldsymbol{x}, \mathsf{y}^+, \mathsf{y}^-) = \eta^+ \left( \mathbb{E}_{\epsilon, t}\left[ s_{\boldsymbol{\theta}}(\boldsymbol{x}, \mathsf{y}^+) \right] - \Delta^+ \right) - \eta^- \left( \mathbb{E}_{\epsilon, t}\left[ s_{\boldsymbol{\theta}}(\boldsymbol{x}, \mathsf{y}^-) \right] - \Delta^- \right). \tag{8}$$

The Circle loss degenerates to AM-Softmax $\mathcal{L}_{\mathrm{AM}}$ loss [53], an important variant of the binary cross-entropy loss 7, when $\iota(\boldsymbol{x}, \mathsf{y}^+, \mathsf{y}^-)$ is defined as:

$$\iota(\boldsymbol{x}, \mathsf{y}^+, \mathsf{y}^-) = \mathbb{E}_{\epsilon, t}\left[ s_{\boldsymbol{\theta}}(\boldsymbol{x}, \mathsf{y}^+) \right] - \left( \mathbb{E}_{\epsilon, t}\left[ s_{\boldsymbol{\theta}}(\boldsymbol{x}, \mathsf{y}^-) \right] - \Delta^- \right). \tag{9}$$

## 4  ABC for Diffusion Models

In this section, we introduce our Alignment by Classification (ABC) framework for diffusion models. We first reformulate alignment as a classification task. To mitigate the instability from the semi-supervised nature of preference data, we apply data augmentation to enable supervised learning. Finally, we present the ABC objective for preference alignment.

## 4.1 The Connection Between Alignment and Classification

In the following sections, we assume each text prompt corresponds to a single aligned image. Let $\boldsymbol{x}_\mathsf{y}^+$ denote the image aligned with prompt $\mathsf{y}$, and $\boldsymbol{x}_\mathsf{y}^-$ a misaligned one. Here, we provide a classification perspective on diffusion model preference alignment [51], formalized through two theorems.

The first theorem shows that the Diffusion-DPO loss serves as an upper bound on the diffusion classification score (6). Specifically, the Diffusion-DPO loss [51] is defined as Equation (10), where $s_{\mathrm{ref}}(\boldsymbol{x}, \mathsf{y}) = \omega_t \|\boldsymbol{\epsilon} - \boldsymbol{\epsilon}_{\mathrm{ref}}(\sqrt{\alpha_t}\boldsymbol{x} + \sigma_t\boldsymbol{\epsilon}, \mathsf{y}, t)\|_2^2$, and $\boldsymbol{\epsilon}_{\mathrm{ref}}$ is a reference diffusion model. Therefore, minimizing this loss corresponds to training a diffusion-based classifier.

$$\mathcal{L}_{\mathrm{DDPO}} = \mathbb{E}_{\epsilon,t}\left[\log\left(1 + \exp\left(-\left(s_{\boldsymbol{\theta}}(\boldsymbol{x}_\mathsf{y}^-, \mathsf{y}) - s_{\mathrm{ref}}(\boldsymbol{x}_\mathsf{y}^-, \mathsf{y})\right)\right)\exp\left(s_{\boldsymbol{\theta}}(\boldsymbol{x}_\mathsf{y}^+, \mathsf{y}) - s_{\mathrm{ref}}(\boldsymbol{x}_\mathsf{y}^+, \mathsf{y})\right)\right)\right], \quad (10)$$

Unlike common practices where $\boldsymbol{\epsilon}_{\mathrm{ref}}(\boldsymbol{x}_t, \mathsf{y}, t)$ is set as the SFT checkpoint [40], we consider it as the ideal model here for discussion. The reason is that using the ideal alignment model as the reference model should produce optimal performance in preference optimization. Thus, it deserves a typical case to discuss in the following theorem.

**Theorem 1.** *(Proof in the supplementary material) We say a diffusion model $\boldsymbol{\epsilon}_{\mathrm{ali}}(\boldsymbol{x}_t, \mathsf{y}, t)$ is ideal alignment if it satisfies $\|\boldsymbol{\epsilon}_{\mathrm{ali}}(\boldsymbol{x}_{t;\mathsf{y}}^+, \mathsf{y}, t) - \boldsymbol{\epsilon}\|_2^2 = 0$ and $\|\boldsymbol{\epsilon}_{\mathrm{ali}}(\boldsymbol{x}_{t;\mathsf{y}}^-, \mathsf{y}, t) - \boldsymbol{\epsilon}\|_2^2 = \delta$ for any $\mathsf{y}$. Here, $\boldsymbol{x}_{t;\mathsf{y}}^+ = \sqrt{\alpha_t}\boldsymbol{x}_\mathsf{y}^+ + \sigma_t\boldsymbol{\epsilon}$ and $\boldsymbol{x}_{t;\mathsf{y}}^- = \sqrt{\alpha_t}\boldsymbol{x}_\mathsf{y}^- + \sigma_t\boldsymbol{\epsilon}$. When the reference model $\boldsymbol{\epsilon}_{\mathrm{ref}}(\boldsymbol{x}_t, \mathsf{y}, t) = \boldsymbol{\epsilon}_{\mathrm{ali}}(\boldsymbol{x}_t, \mathsf{y}, t)$ in Equation (10) is an ideal alignment model and $s_{\mathrm{ali}}(\boldsymbol{x}, \mathsf{y}) = w_t\|\boldsymbol{\epsilon} - \boldsymbol{\epsilon}_{\mathrm{ali}}(\sqrt{\alpha_t}\boldsymbol{x} + \sigma_t\boldsymbol{\epsilon}, \mathsf{y}, t)\|_2^2$, the AM-Softmax loss (9) is upper bounded by the Diffusion-DPO loss (10). Specifically, we have*

$$\log\left(1 + \exp\left(-\left(\mathbb{E}_{\epsilon,t}\left[s_{\boldsymbol{\theta}}(\boldsymbol{x}_\mathsf{y}^-, \mathsf{y})\right] - \delta\right)\right)\right)\exp\left(\mathbb{E}_{\epsilon,t}\left[s_{\boldsymbol{\theta}}(\boldsymbol{x}_\mathsf{y}^+, \mathsf{y})\right]\right)$$
$$\leq \mathbb{E}_{\epsilon,t}\left[\log\left(1 + \exp\left(-\left(s_{\boldsymbol{\theta}}(\boldsymbol{x}_\mathsf{y}^-, \mathsf{y}) - s_{\mathrm{ali}}(\boldsymbol{x}_\mathsf{y}^-, \mathsf{y})\right)\right)\exp\left(s_{\boldsymbol{\theta}}(\boldsymbol{x}_\mathsf{y}^+, \mathsf{y}) - s_{\mathrm{ali}}(\boldsymbol{x}_\mathsf{y}^+, \mathsf{y})\right)\right)\right]. \quad (11)$$

This theorem indicates that the Diffusion-DPO loss serves as an upper bound for the AM-Softmax loss [53]. Thus, minimizing the Diffusion-DPO loss with an ideal reference model will also minimize the AM-Softmax loss. This implies that performing the alignment task results in performing the classification task for diffusion models.

The second theorem shows that the predicted noise in a diffusion model is a weighted average of noise estimates across all possible images. To generate an image aligned with a prompt, the model must increase the weight on the noise corresponding to the aligned image—highlighting that strong classification ability is essential for alignment.

**Theorem 2.** *(Proof in the supplementary material) Let $\mathsf{Y} = \{\mathsf{y}_i\}_{i=1}^N$ denote $N$ text prompts and $D = \{\boldsymbol{x}_{\mathsf{y}_i}\}$ be corresponding aligned images. We assume that the prior prompt distribution $p(\mathsf{y})$ and image distribution $p(\boldsymbol{x})$ are uniform. To describe the discriminative ability, we define the conditional probability $p(\boldsymbol{x}|\mathsf{y})$ as*

$$p(\mathsf{y}|\boldsymbol{x}_{\mathsf{y}_i}) = \begin{cases} \frac{n}{N} & \mathsf{y} = \mathsf{y}_i, \\ \frac{N-n}{N(N-1)} & \mathsf{y} \in \mathsf{Y} - \{\mathsf{y}_i\}. \end{cases} \quad \text{where } n < N. \quad (12)$$

*Then, $p(\boldsymbol{x}|\mathsf{y}) = p(\mathsf{y}|\boldsymbol{x})$ and the optimal diffusion model $\boldsymbol{\epsilon}_{\mathrm{opt}}(\boldsymbol{x}_t, \mathsf{y}, t)$, which achieves minimal diffusion loss over both the training set and the test set, over $D$ is given by:*

$$\boldsymbol{\epsilon}_{\mathrm{opt}}(\boldsymbol{x}_t, \mathsf{y}, t) = \sum_{\boldsymbol{x}^{(i)} \in D} \frac{w_i}{\sum_{\boldsymbol{x}^{(j)} \in D} w_j} \cdot \boldsymbol{\epsilon}_i, \quad (13)$$

*where $\boldsymbol{\epsilon}_i = \frac{\boldsymbol{x}_t - \sqrt{\alpha_t}\boldsymbol{x}^{(i)}}{\sigma_t}$, $\lambda = \frac{n(N-1)}{N-n}$, $w_i = \begin{cases} \lambda \cdot \exp\left(-\frac{\|\boldsymbol{x}_t - \sqrt{\alpha_t}\boldsymbol{x}^{(i)}\|_2^2}{2\sigma_t^2}\right), & \boldsymbol{x}^{(i)} \in \{\boldsymbol{x}_\mathsf{y}\}, \\ \exp\left(-\frac{\|\boldsymbol{x}_t - \sqrt{\alpha_t}\boldsymbol{x}^{(i)}\|_2^2}{2\sigma_t^2}\right), & \boldsymbol{x}^{(i)} \in D - \{\boldsymbol{x}_\mathsf{y}\}. \end{cases}$*

Theorem 2 indicates that the predicted noise of the optimal diffusion model is the weighted average of the noise $\boldsymbol{\epsilon}_i$, which denotes the exact noise contained in $\boldsymbol{x}_t$ with respect to the clean image $\boldsymbol{x}^{(i)} \in D$. In order to control the diffusion model to approximate the image $\boldsymbol{x}_\mathsf{y}$ that is aligned with the text $\mathsf{y}$, it has to make the predicted noise approximate to the noise $\frac{\boldsymbol{x}_t - \sqrt{\alpha_t}\boldsymbol{x}_\mathsf{y}}{\sigma_t}$. This further leads $n$ to approximate $N$, which means we maximize $p(\mathsf{y}|\boldsymbol{x}_\mathsf{y})$. Since diffusion models provide a good estimation for $p(\mathsf{y}|\boldsymbol{x})$ according to Equation (4), this implies that once the diffusion model is an ideal classifier, the diffusion model $\boldsymbol{\epsilon}_{\mathrm{opt}}(\boldsymbol{x}_t, \mathsf{y}, t)$ will be an ideal alignment model.

Finally, to enhance understanding, we briefly interpret the two theorems. Theorem 1 proves that the AM-Softmax loss is upper bounded by the Diffusion-DPO loss. In other words, minimizing the Diffusion-DPO loss for better alignment will also reduce the AM-Softmax loss, leading to improved classification performance. Simply put, better alignment leads to better classification. Conversely, Theorem 2 shows that, under certain conditions, improved classification leads to better alignment. Together, these two theorems reveal a strong connection between classification and alignment, forming the theoretical foundation of our approach, which replaces the DPO loss with the ABC loss for alignment tasks. We first establish a connection between alignment and classification.

## 4.2 The Connection Between Alignment and Semi-Supervised Learning

Diffusion model alignment is a form of semi-supervised learning using a human preferences dataset. Specifically, Diffusion-DPO [51] is fine-tuned on Pick-a-Pic [23], a human preference dataset for text-to-image generation. To construct the dataset, an SFT model generates pairs of images $(\boldsymbol{x}_1, \boldsymbol{x}_2)$ from a given prompt y. These pairs are then shown to human annotators, who indicate a preference, denoted as $\boldsymbol{x}_\mathsf{y}^+ \succ \boldsymbol{x}_\mathsf{y}^-$, where $\boldsymbol{x}_\mathsf{y}^+$ and $\boldsymbol{x}_\mathsf{y}^-$ are the preferred and dispreferred images, respectively. Each dataset item is thus a triplet $(\mathsf{y}, \boldsymbol{x}_\mathsf{y}^+, \boldsymbol{x}_\mathsf{y}^-)$. In this setup, y serves as the correct label for $\boldsymbol{x}_\mathsf{y}^+$, while $\boldsymbol{x}_\mathsf{y}^-$ lacks a corresponding optimal prompt. As alignment resembles a classification task in which only half the data has labels, it naturally fits within a semi-supervised classification framework.

Regularization is critical for stable semi-supervised training. When aligning the diffusion model using AM-Softmax—as suggested in Theorem 1—the optimization problem reduces to:

$$\min_{\boldsymbol{\theta}} \sum_{\mathsf{y}\in\mathsf{Y}} \log\left(1 + \exp\left(\iota(\boldsymbol{x}_\mathsf{y}^-, \boldsymbol{x}_\mathsf{y}^+, \mathsf{y})\right)\right), \text{ where}$$
$$\iota(\boldsymbol{x}^-, \boldsymbol{x}^+, \mathsf{y}) = \mathbb{E}_{\epsilon,t}\left[s_{\boldsymbol{\theta}}(\boldsymbol{x}^+, \mathsf{y})\right] - \left(\mathbb{E}_{\epsilon,t}\left[s_{\boldsymbol{\theta}}(\boldsymbol{x}^-, \mathsf{y})\right] - \Delta^-\right). \tag{14}$$

Since $\boldsymbol{x}_\mathsf{y}^-$ lacks a corresponding prompt, the optimization tends to maximize $s_{\boldsymbol{\theta}}(\boldsymbol{x}_\mathsf{y}^-, \mathsf{y})$, which can lead to $\|\epsilon - \epsilon_{\boldsymbol{\theta}}(\sqrt{\alpha_t}\boldsymbol{x}_\mathsf{y}^- + \sigma_t\epsilon, \mathsf{y}, t)\|_2^2$ becoming arbitrarily large. However, even if $\boldsymbol{x}_\mathsf{y}^-$ is less preferred than $\boldsymbol{x}_\mathsf{y}^+$, this reconstruction error should still remain bounded; otherwise, the diffusion model will lose its ability to generate valid images. A practical compromise to prevent the loss from diverging is to select a large $\Delta^-$, effectively modeling an ideal alignment function with large $\|\epsilon_{\mathrm{ali}}(\boldsymbol{x}_{t;\mathsf{y}}^-, \mathsf{y}, t) - \epsilon\|_2^2$ during DPO optimization. However, this does not fully resolve the issue, which helps explain why Diffusion-DPO may fail to reliably train diffusion models in some cases.

We regularize classification through data augmentation to mitigate instability caused by missing prompts in half of the user preference dataset. Specifically, we define $\mathsf{y}^+$ as the original prompt y, and construct $\mathsf{y}^-$ by appending "The image that aligns less with human preferences" to y. This reformulates each preference tuple $(\mathsf{y}, \boldsymbol{x}_\mathsf{y}^+, \boldsymbol{x}_\mathsf{y}^-)$ into two supervised examples: $(\mathsf{y}^+, \boldsymbol{x}_{\mathsf{y}^+})$ and $(\mathsf{y}^-, \boldsymbol{x}_{\mathsf{y}^-})$. The task thus becomes a binary classification problem between images conditioned on $\mathsf{y}^+$ and $\mathsf{y}^-$, converting the semi-supervised objective (14) into a fully supervised one (15), which helps stabilize training by constraining the residual error term.

$$\min_{\boldsymbol{\theta}} \sum_{\mathsf{y}\in\mathsf{Y}} \left[\log\left(1 + \exp\left(\iota(\boldsymbol{x}_{\mathsf{y}^-}, \boldsymbol{x}_{\mathsf{y}^+}, \mathsf{y}^+)\right)\right) + \log\left(1 + \exp\left(\iota(\boldsymbol{x}_{\mathsf{y}^+}, \boldsymbol{x}_{\mathsf{y}^-}, \mathsf{y}^-)\right)\right)\right], \text{ where}$$
$$\iota(\boldsymbol{x}^-, \boldsymbol{x}^+, \mathsf{y}) = \mathbb{E}_{\epsilon,t}\left[s_{\boldsymbol{\theta}}(\boldsymbol{x}^+, \mathsf{y})\right] - \left(\mathbb{E}_{\epsilon,t}\left[s_{\boldsymbol{\theta}}(\boldsymbol{x}^-, \mathsf{y})\right] - \Delta_\mathsf{y}^-\right). \tag{15}$$

## 4.3 ABC Loss for Alignment

Theorem 1 shows that optimizing the Diffusion-DPO loss (10) effectively minimizes the AM-Softmax loss (9). Theorem 2 further demonstrates that achieving ideal alignment requires the diffusion model to be discriminative. Section 4.2 attributes training instability of alignment to the semi-supervised nature of the task, which we address through a data augmentation strategy that converts it into a supervised classification problem. Together, these results support the feasibility of aligning diffusion models to human preferences via classification.

Alignment by classification imposes an important constraint: the expected score for the less preferred image, $\mathbb{E}_{\epsilon,t}\left[s_{\boldsymbol{\theta}}(\boldsymbol{x}^-, \mathsf{y})\right]$, must be properly bounded. If this value becomes too large, the model is compelled to increase $\mathbb{E}_{\epsilon,t}\left[s_{\boldsymbol{\theta}}(\boldsymbol{x}^+, \mathsf{y})\right]$ accordingly, which may cause the diffusion model to fail in generating coherent images. Conversely, if $\mathbb{E}_{\epsilon,t}\left[s_{\boldsymbol{\theta}}(\boldsymbol{x}^-, \mathsf{y})\right]$ is too small, the model becomes

insufficiently discriminative, weakening its ability to align with human preferences according to Theorem 2. Ideally, the model should maintain a margin-based separation:

$$\mathbb{E}_{\epsilon,t}\left[s_{\boldsymbol{\theta}}(\boldsymbol{x}^-,\mathsf{y})\right] = \mathbb{E}_{\epsilon,t}\left[s_{\boldsymbol{\theta}}(\boldsymbol{x}^+,\mathsf{y})\right] + \delta, \tag{16}$$

where $\delta > 0$ is a fixed positive margin that ensures both discriminability and stability. To enforce this constraint, we adopt the Circle loss (8)—a generalized version of the AM-Softmax loss—which better accommodates the margin-based formulation. Specifically, we introduce the Alignment by Circle (ABC) loss, denoted as $\mathcal{L}_{\text{ABC}}$, defined as follows:

$$\mathcal{L}_{\text{ABC}}(\boldsymbol{\theta}) = \sum\nolimits_{\mathsf{y}\in\mathsf{Y}}\left[\log\left(1+\exp\left(\iota(\boldsymbol{x}_{\mathsf{y}^-},\boldsymbol{x}_{\mathsf{y}^+},\mathsf{y}^+)\right)\right) + \log\left(1+\exp\left(\iota(\boldsymbol{x}_{\mathsf{y}^+},\boldsymbol{x}_{\mathsf{y}^-},\mathsf{y}^-)\right)\right)\right], \text{ where}$$

$$\iota(\boldsymbol{x}^-,\boldsymbol{x}^+,\mathsf{y}) = \eta_{\mathsf{y}}^+\left(\mathbb{E}_{\epsilon,t}\left[s_{\boldsymbol{\theta}}(\boldsymbol{x}^+,\mathsf{y})\right]-\Delta_{\mathsf{y}}^+\right) - \eta_{\mathsf{y}}^-\left(\mathbb{E}_{\epsilon,t}\left[s_{\boldsymbol{\theta}}(\boldsymbol{x}^-,\mathsf{y})\right]-\Delta_{\mathsf{y}}^-\right). \tag{17}$$

Here, $\eta_{\mathsf{y}}^+$ and $\eta_{\mathsf{y}}^-$ act as self-paced weighting factors that adaptively emphasize samples with suboptimal scores—specifically, those far from their ideal values $O_{\mathsf{y}}^+$ and $O_{\mathsf{y}}^-$—to ensure stronger gradients and more effective updates. We define these weights as follows:

$$\begin{cases}\eta_{\mathsf{y}}^+ = \mathbb{E}_{\epsilon,t}\left[s_{\boldsymbol{\theta}}(\boldsymbol{x}^+,\mathsf{y})\right]-O_{\mathsf{y}}^+, \\ \eta_{\mathsf{y}}^- = O_{\mathsf{y}}^- - \mathbb{E}_{\epsilon,t}\left[s_{\boldsymbol{\theta}}(\boldsymbol{x}^-,\mathsf{y})\right].\end{cases} \begin{cases}O_{\mathsf{y}}^+ = 0, \\ \Delta_{\mathsf{y}}^+ = 0.\end{cases} \begin{cases}O_{\mathsf{y}}^- = \text{sg}[\mathbb{E}_{\epsilon,t}[s_{\boldsymbol{\theta}}(\boldsymbol{x}_{\mathsf{y}}^+,\mathsf{y})]] + \delta, \\ \Delta_{\mathsf{y}}^- = \text{sg}[\mathbb{E}_{\epsilon,t}[s_{\boldsymbol{\theta}}(\boldsymbol{x}_{\mathsf{y}}^+,\mathsf{y})]] + \delta.\end{cases} \tag{18}$$

Since $\log\left(1+\exp(x)\right)$ is a monotonically increasing function, minimizing the loss is equivalent to minimizing the term $\iota(\boldsymbol{x}^-,\boldsymbol{x}^+,\mathsf{y})$. Substituting Equation (18) into this term yields:

$$\iota(\boldsymbol{x}^-,\boldsymbol{x}^+,\mathsf{y}) = \left(\mathbb{E}_{\epsilon,t}\left[s_{\boldsymbol{\theta}}(\boldsymbol{x}^+,\mathsf{y})\right]-\frac{O_{\mathsf{y}}^++\Delta_{\mathsf{y}}^+}{2}\right)^2 + \left(\mathbb{E}_{\epsilon,t}\left[s_{\boldsymbol{\theta}}(\boldsymbol{x}^-,\mathsf{y})\right]-\frac{O_{\mathsf{y}}^-+\Delta_{\mathsf{y}}^-}{2}\right)^2, \tag{19}$$

where $\text{sg}[\cdot]$ denotes stop-gradient (i.e., the value is detached from backpropagation), and $\delta$ introduces a soft margin to ensure a separation between positive and negative scores. The minimizer of this objective can be verified as $(0,\text{sg}[\mathbb{E}_{\epsilon,t}[s_{\boldsymbol{\theta}}(\boldsymbol{x}_{\mathsf{y}}^+,\mathsf{y})]] + \delta)$. Accordingly, the loss drives $\mathbb{E}_{\epsilon,t}\left[s_{\boldsymbol{\theta}}(\boldsymbol{x}^+,\mathsf{y})\right]$ toward 0, while encouraging $\mathbb{E}_{\epsilon,t}\left[s_{\boldsymbol{\theta}}(\boldsymbol{x}^-,\mathsf{y})\right]$ to approach $\text{sg}[\mathbb{E}_{\epsilon,t}[s_{\boldsymbol{\theta}}(\boldsymbol{x}_{\mathsf{y}}^+,\mathsf{y})]] + \delta$—precisely the behavior we desire for effective alignment.

## 5 Experiments

We present both qualitative and quantitative experiments in Section 5.1 to highlight the alignment advantages of our method. In Section 5.2, we provide a deeper analysis of why our approach outperforms existing diffusion model alignment techniques. In Section 5.3, we conduct an ablation study to examine how the hyperparameters in the ABC loss affect alignment quality.

### 5.1 Human Preference Alignment Comparison

We present both quantitative and qualitative comparisons for human preference alignment. Our approach builds on the Diffusion-DPO codebase [51]. We train the models using the AdamW [30] optimizer for SD1.5, Adafactor [45] optimizer for SDXL on 8 A6000 GPUs, with a batch size of 2, gradient accumulation of 128 steps and a learning rate of $1 \times 10^{-8}$, incorporating a linear warmup schedule. For SD1.5 and SDXL training, $\delta$ is set to 0.025. These settings largely follow the original Diffusion-DPO configuration, with minor modifications to enhance training efficiency. We apply our proposed ABC loss to fine-tune both the SD1.5 and SDXL base models, resulting in our SD1.5-ABC and SDXL-ABC variants. For clarity, we adopt a "Model–Method" naming convention—e.g., SDXL-ABC refers to the SDXL model fine-tuned with the ABC loss.

#### 5.1.1 Qualitative Comparison

We present a qualitative comparison of SDXL-ABC with SDXL-Base, SDXL-DPO [51], SDXL-SPO [27], and SDXL-MAPO [20] in Figure 1, where SDXL-Base represents the original SDXL-1.0 model. As shown in Figure 1, SDXL-ABC generates images with clear improvements in text-image alignment. To assist readers in identifying mismatches between the text and the generated images, we highlight relevant textual phrases in color and enclose the corresponding image regions in bounding boxes. Our method incorporates human preferences through direct optimization based on user feedback, resulting in more engaging visuals, such as vivid color palettes, dramatic lighting, coherent compositions, fine detail, creative elements, consistent color harmony, and structured multi-object arrangements. More important, the generated text-image pairs are also more semantically aligned.

| Prompts | SDXL-Base | SDXL-DPO | SDXL-SPO | SDXL-MAPO | SDXL-ABC |
|---|---|---|---|---|---|

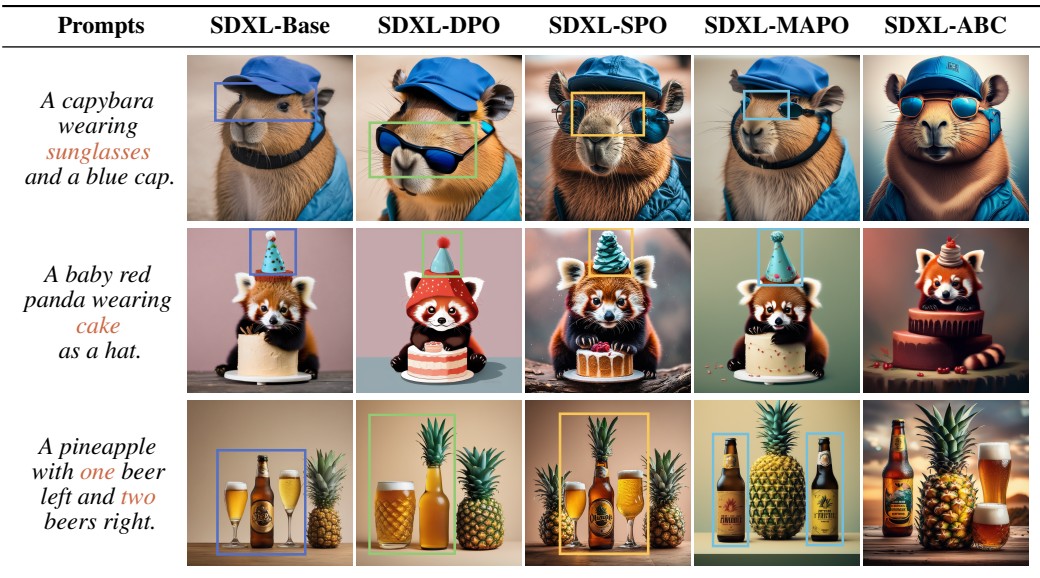

Figure 1: Qualitative Comparison for Diffusion Model Alignment. We develop an alignment-by-classification approach to align diffusion models with human preferences. Fine-tuned from the SDXL-1.0 model, our method generates images with improved visual appeal and textual alignment compared to other alignment baselines. In our comparisons, SDXL-DPO [51], SDXL-SPO [27] and SDXL-MAPO [20] denote competing aligned variants and SDXL-Base denote the SDXL-1.0 model.

Table 1: Quantitative Win-rate Comparison Using Automated Preference Metrics. We evaluate the alignment performance of diffusion models using prompts from HPS and PartiPrompts across various evaluators. Both SDXL and SD1.5 serve as base models. Win rates above 50%—indicating superior performance over the baseline—are highlighted in bold. We note that MAPO has not released their SD1.5-based checkpoint, and KTO has not released their SDXL-based checkpoint.

| | PartiPrompts | | | | HPS benchmark | | | |
|---|---|---|---|---|---|---|---|---|
| | PickScore | HPS | Aesthetics | CLIP | PickScore | HPS | Aesthetics | CLIP |
| vs. SD1.5-Base | **60.02** | **81.51** | **74.27** | **59.72** | **74.83** | **85.75** | **68.84** | **59.65** |
| vs. SD1.5-DPO | **55.85** | **73.02** | **64.90** | 44.97 | **53.46** | **71.50** | **64.19** | **52.06** |
| vs. SD1.5-SPO | **51.16** | **61.59** | 47.60 | **60.02** | 45.35 | **54.99** | 38.08 | **64.83** |
| vs. SD1.5-KTO | **57.77** | 44.72 | **53.90** | 47.22 | **52.28** | 42.88 | **52.86** | **53.93** |
| vs. SDXL-Base | **74.38** | **79.26** | **80.20** | **52.46** | **79.35** | **70.17** | **72.28** | **60.38** |
| vs. SDXL-DPO | **73.22** | **72.50** | **68.25** | **50.51** | **77.26** | **69.54** | **70.19** | **57.06** |
| vs. SDXL-SPO | **52.49** | 40.31 | **59.93** | **55.53** | **51.16** | **52.41** | 46.78 | **59.87** |
| vs. SDXL-MAPO | **65.35** | **81.17** | **72.10** | 46.97 | **68.55** | **64.89** | **68.18** | **51.14** |

### 5.1.2 Quantitative Comparison

We compare our method against existing baselines—including SD1.5, SDXL, and their DPO, SPO, KTO, and MAPO variants—using both user studies and automated preference metrics. For automated evaluation, we assess Pick Score [23], HPS [58], LAION Aesthetics [44], and CLIP [39], using prompts from the HPS benchmark [58] and PartiPrompts [62]. We report win rates between our method and each baseline under these metrics in Table 1, while Table 2 presents results on the GenEval benchmark evaluating model performance across 8,000 prompts.

To confirm the method's efficacy, we conducted a user study. Specifically, we randomly sampled 100 prompts from the PartiPrompts dataset and another 100 prompts from the HPSv2 benchmark. For each prompt, we generated five images using five different methods. Participants were shown five images per prompt (one from each method) and asked to answer three questions: Q1 *Which image is your overall preferred choice?* Q2 *Which image is more visually attractive?* Q3 *Which image better matches the text description?* To minimize position bias, the order of images was randomized for

Table 2: Quantitative comparison on GenEval. We evaluate model performance on 8,000 prompts spanning attribute binding, relationships, numeracy, and complex compositions. Higher scores indicate stronger alignment with the intended composition.

| Model | Color (B-VQA) | Shape (B-VQA) | Texture (B-VQA) | Numeracy (UniDet) | 2D-Spatial (UniDet) | 3D-Spatial (UniDet) | Non-Spatial (CLIP) | Complex (3-in-1) |
|---|---|---|---|---|---|---|---|---|
| SD1.5-Base | 0.3811 | 0.3395 | 0.4192 | 0.4436 | 0.1460 | 0.2912 | 0.3092 | 0.3002 |
| SD1.5-DPO | 0.3943 | 0.3440 | 0.4374 | 0.4523 | 0.1627 | 0.3090 | 0.3091 | 0.3032 |
| SD1.5-SPO | 0.4030 | 0.4001 | 0.4152 | 0.4461 | 0.1471 | 0.2958 | 0.3010 | 0.3131 |
| SD1.5-KTO | 0.4645 | 0.3815 | 0.4730 | 0.4618 | 0.1919 | 0.3318 | 0.3104 | 0.3514 |
| **SD1.5-ABC** | **0.4647** | **0.4005** | **0.4751** | **0.4570** | **0.1895** | **0.3324** | **0.3106** | **0.3587** |
| SDXL-Base | 0.5708 | 0.4880 | 0.5600 | 0.5591 | 0.1949 | 0.3551 | 0.3065 | 0.4383 |
| SDXL-DPO | 0.6586 | 0.5358 | 0.6521 | 0.5300 | 0.2376 | 0.3668 | 0.3116 | 0.4923 |
| SDXL-SPO | 0.6431 | 0.5200 | 0.6496 | 0.5765 | 0.2298 | 0.3513 | 0.3031 | 0.4424 |
| SDXL-MAPO | 0.6682 | 0.5104 | 0.5650 | 0.5189 | 0.1700 | 0.3507 | 0.3136 | 0.4401 |
| **SDXL-ABC** | **0.6708** | **0.5450** | **0.6866** | **0.5623** | **0.2401** | **0.3697** | **0.3154** | **0.5051** |

Figure 2: Quantitative Win-rate Comparison Using User Study. Both SD1.5-ABC and SDXL-ABC outperform the baselines, with the top figure showing results on PartiPrompts and the bottom showing results on the HPS benchmark. The rows for SD1.5 and SDXL indicate that the base diffusion models are SD1.5 and SDXL, respectively. Our method consistently generates outputs with higher overall preference across two key dimensions: visual appeal and prompt alignment. We note that MAPO has not released their SD1.5-based checkpoint, and KTO has not released their SDXL-based checkpoint.

each prompt. Each method's final score was computed as a weighted sum of its win rates under the three criteria, with weights of 30% for general preference, 30% for visual appeal, and 40% for prompt alignment. The study was conducted as a blind evaluation. Annotators were not informed about which method generated each image. We recruited participants from our research group, comprising approximately 100 students, and collected a total of 82 valid responses.

Table 1 reports the win rates of ABC-aligned diffusion models against their respective baselines. Fine-tuning with our ABC loss consistently improves performance for both SD1.5 and SDXL across nearly all metrics and datasets, demonstrating the effectiveness of our approach. Table 2 shows the quantitative comparison on GenEval, with ABC achieving competitive or superior performance across various compositional tasks. Figure 2 further illustrates user study results, where our method receives the highest number of winning votes in terms of general preference and visual appeal. For instance, on the HPS dataset with SDXL, ABC achieves a leading win rate of 36.5% in general preference among five competing methods.

## 5.2 Performance Analysis

Theorem 1 demonstrates that enhancing a model's alignment capability improves its classification performance, while Theorem 2 indicates that stronger discriminative ability leads to better alignment. In this section, we evaluate the zero-shot classification performance of aligned diffusion models to investigate the intrinsic connection between alignment and classification. We also analyze their discriminative strength, providing experimental evidence to support the effectiveness of our approach.

### 5.2.1 Zero-Shot Classification

To validate the effectiveness of Theorem 1, we evaluate zero-shot classification performance on six benchmark datasets: Food-101 [3], CIFAR-10 [24], Aircraft [32], Pets [36], Flowers102 [33], and STL-10 [10]. We adopt prompt templates and class labels from [39], including refinements to disambiguate class names (e.g., "crane" → "crane bird") [35]. As shown in Table 3, diffusion classifiers

Table 3: **Zero-Shot Classification Performance**. We adopt the Robust Classification via a Single Diffusion Model method [6] to evaluate the classification ability of checkpoints produced by different alignment methods. The results suggest that alignment generally improves the classification capabilities of diffusion models. Among all alignment approaches, our method achieves the highest classification accuracy.

|  | Food-101 | CIFAR-10 | Aircraft | Pets | Flowers102 | STL-10 |
|---|---|---|---|---|---|---|
| SD1.5-Base | 75.87 | 83.15 | 25.47 | 83.53 | 50.33 | 89.44 |
| SD1.5-DPO | 78.57 | 84.79 | 28.39 | 87.37 | 52.96 | 92.38 |
| SD1.5-SPO | 76.04 | 84.59 | 27.25 | 84.64 | 51.94 | 91.06 |
| SD1.5-KTO | 77.93 | 83.92 | 26.71 | 86.21 | 51.13 | 92.03 |
| **SD1.5-ABC (Ours)** | **79.12** | **85.13** | **29.11** | **88.48** | **53.42** | **93.75** |

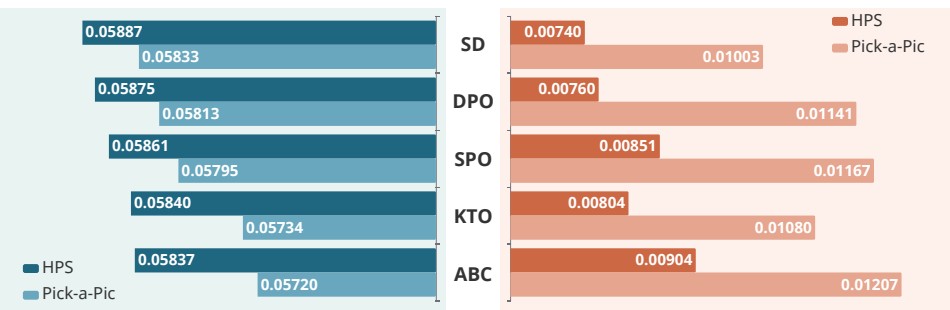

Figure 3: Comparison of Noise Prediction Errors on Pick-a-Pic and HPS. Left: Average noise prediction error $\mathbb{E}_{\epsilon,t}\left[s_{\theta}(\boldsymbol{x}^{+},\mathsf{y})\right]$ for the preferred text-image pair $(\boldsymbol{x}^{+},\mathsf{y})$. Right: Average margin $\Delta = \mathbb{E}_{\epsilon,t}\left[s_{\theta}(\boldsymbol{x}^{-},\mathsf{y})\right] - \mathbb{E}_{\epsilon,t}\left[s_{\theta}(\boldsymbol{x}^{+},\mathsf{y})\right]$ between the noise prediction errors of the preferred pair $(\boldsymbol{x}^{+},\mathsf{y})$ and the dispreferred pair $(\boldsymbol{x}^{-},\mathsf{y})$. A lower noise prediction error suggests higher image quality, while a larger $\Delta$ indicates better discrimination aligned with user preference.

built on aligned models—SD1.5-DPO, SD1.5-SPO, and SD1.5-KTO—consistently outperform the baseline classifier based on the original SD1.5. We attribute this improvement to the fact that the diffusion DPO loss serves as an upper bound to the AM-Softmax classification loss, thereby enhancing alignment with discriminative objectives. These results empirically support the theoretical insight from Theorem 1, which states that better alignment capability improves classification performance. Furthermore, our method outperforms other diffusion-based classifiers, likely due to the explicit use of classification loss to guide alignment.

### 5.2.2 Discriminative Strength Measured by Prediction Error

Theorem 2 confirms that stronger discriminative ability in diffusion models leads to better alignment. Assuming a uniform prompt distribution $p(\mathsf{y})$, Equations (4) and (5) show that the class probability $p(\mathsf{y}|\boldsymbol{x})$ is proportional to $\exp\left(-\mathbb{E}_{\epsilon,t}\left[s_{\theta}(\boldsymbol{x},\mathsf{y})\right]\right)$. Therefore, a lower prediction error indicates better generation quality and a higher likelihood that $\boldsymbol{x}$ belongs to class $\mathsf{y}$. To further improve discriminative power, it is crucial to maximize the margin between prediction errors of positive and negative samples. However, directly maximizing $\exp\left(-\mathbb{E}_{\epsilon,t}\left[s_{\theta}(\boldsymbol{x}_{\mathsf{y}}^{-},\mathsf{y})\right]\right)$ can destabilize training when the prediction error is large. A more stable approach is to maximize the margin $\Delta = \mathbb{E}_{\epsilon,t}\left[s_{\theta}(\boldsymbol{x}^{-},\mathsf{y})\right] - \mathbb{E}_{\epsilon,t}\left[s_{\theta}(\boldsymbol{x}^{+},\mathsf{y})\right]$ ensuring it is large while keeping the generation quality high. Ideally, the noise prediction error $\mathbb{E}_{\epsilon,t}\left[s_{\theta}(\boldsymbol{x}^{+},\mathsf{y})\right]$ should be as low as possible for preferred pairs, and the margin $\Delta$ should be as large as possible to reinforce alignment with human preferences. As shown in Figure 3, our method achieves the lowest prediction error and the largest margin on both the Pick-a-Pic and HPS datasets.

### 5.3 Ablation Study

The ABC loss (17) introduces a hyperparameter $\delta$, which defines the separation margin between preferred and dispreferred samples. This margin directly affects the strength of the alignment signal

during training, consequently, the quality of the generated results. As shown in Table 4, all metrics follow a consistent U-shaped trend, with $\delta = 0.025$ achieving the best overall performance.

When $\delta$ is too small, the model struggles to distinguish between preferred and dispreferred outputs. This often leads to trivial solutions—such as degrading dispreferred samples to minimize the loss—which ultimately harms both generation fidelity and alignment quality (see the first row of the table). Conversely, when $\delta$ is too large, the model enforces an overly strict separation, resulting in

Table 4: Ablation Study on $\delta$, which controls the discrimination strength of the diffusion model in the ABC objective. $\delta$=0.025 yields the best performance.

| $\delta$ | PickScore ↑ | HPS ↑ | Aesthetics ↑ | CLIP ↑ |
|---|---|---|---|---|
| 0.005 | 18.34 | 24.25 | 4.55 | 22.16 |
| 0.015 | 19.97 | 25.72 | 4.96 | 28.79 |
| **0.025** | **21.79** | **27.67** | **5.65** | **33.86** |
| 0.035 | 20.08 | 26.09 | 4.61 | 29.81 |
| 0.055 | 14.27 | 19.75 | 3.32 | 8.87 |

high noise prediction errors for dispreferred samples, which negatively impacts generation quality (as shown in the last row). In summary, a small $\delta$ limits the discriminative power of the model, weakening alignment, while a large $\delta$ increases prediction error, degrading output quality.

## 6 Conclusion

In this work, we propose a method for aligning text-to-image diffusion models with human preferences through classification. We begin by reformulating the alignment task as a classification problem, showing that optimizing the Diffusion-DPO loss effectively minimizes the AM-Softmax loss, and demonstrating that achieving ideal alignment requires the diffusion model to be discriminative. Building on this insight, we introduce the Alignment by Circle (ABC) loss to guide diffusion models toward human-aligned outputs. From the classification perspective, we identify that human preference datasets are inherently semi-supervised and propose a data augmentation strategy to convert them into fully supervised datasets for more stable ABC training. Experimental results show that our method outperforms previous approaches in human preference alignment, highlighting the effectiveness of alignment by classification for fine-tuning diffusion-based text-to-image models.

## 7 Limitation

In this paper, we reveal the connection between discriminative ability and alignment performance through Theorem 1 and 2. However, this connection remains primarily qualitative. A promising direction for future research is to establish a quantitative relationship between discriminative strength and alignment effectiveness. Moreover, since preference data are inherently noisy, it is often challenging to define a clear criterion for determining whether one image is preferred over another. Although we do not explicitly address this issue in the current work, our formulation offers a potential path forward: by transforming the alignment task into a classification problem, we can leverage the extensive literature on classification under label noise. Adapting these techniques to noisy alignment scenarios may provide a principled solution, which we leave for future investigation.

## Acknowledgment

This work was supported by the National Natural Science Foundation of China (62372237,62332010) and the Major Science and Technology Projects in Jiangsu Province under Grant BG2024042.

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
