# OpenReview forum: "Aligning Text-to-Image Diffusion Models to Human Preference by Classification"
_NeurIPS.cc/2025/Conference — NeurIPS 2025 spotlight_

### Official Review · Reviewer_RW4E · 2025-06-20

**Clarity:** 4
**Significance:** 3
**Originality:** 3
**Rating:** 4
**Confidence:** 4

**Summary:**

This work deals with the alignment of text-to-image generative models. In short, diffusion-based generative models (and other generative models such as flow matching, rectified flows, and so on, by the way) suffer from a number of problems when generating an image, given a prompt. Those issues range from catastrophic neglecting (objects present in the prompt are not generated as part of the final image), attribute binding (textures, colors, and other image characteristics) are not properly respected, spatial binding (positioning of objects as defined by the prompt) is not respected, numbering (e.g. object counts) is not correct, and many more.

The authors build on fine-tuning methods that were originally devised for LLM alignment, and that have been also recently applied to diffusion models, to derive a new approach that relates alignment to classification accuracy. They amend the existing connection between alignment and semi-supervised learning by introducing a simple form of data augmentation, transforming the problem into a supervised classification task. Leveraging insights on bounds on the AM-Softmax loss and the Diffusion-DPO loss, as well as results indicating that ideal alignment requires the diffusion model to be discriminative, they propose a new loss that is an instance of the Circle Loss, which is used to fine-tune the (conditional) score network based on positive and negative examples, with their associated labels, plus additional parameters to improve the discriminative power of the diffusion model.

Experiments using standard metrics and a user study indicate that the proposed ABC-loss works well and outperforms existing variants of DPO-based alignment methods. Additional experiments serve the purpose of validating the theoretical analysis, and study the impact of one important hyper-parameter of the proposed method.

**Questions:**

* If recent autoregressive generative models are becoming the new standard for image generation, how relevant do you consider alignment approaches for diffusion models in general, and DPO-based fine-tuning approaches in particular, will be?

* What are the key advantages of DPO-based alignment methods such as ABC when compared to other fine-tuning methods from the literature (e.g., you cite DPOK as reference [14], but do not even refer to it in the main), or other inference-time methods?

* How would your approach perform with more natural prompts than the ones you used in your evaluation? Would it be possible to consider using T2I-CompBench++ (reference [2] above) for your evaluation?

* Could you report absolute metrics rather than percentage of wins in Table 1?

* In sec 4.1, line 116, you rely on a pretty strong assumption. I realize your work is not the only one requiring such an assumption, but it would be interesting to hear your opinion on how a DPO-based framework such as ABC could cope with the more realistic case of multiple images being aligned to a single prompt. Consider a realistic user prompt: natural language is ambiguous, subject to interpretation, thus there might be multiple generated images that could be considered as aligned to it

* In Theorem 2, lines 142-143, you need to assume $p(y)$ and $p(x)$ to be uniform distributions. Also this assumption is pretty strong. Can you comment on what would it be necessary to do if we relax such an assumption?

**Ethical Concerns:**

["NO or VERY MINOR ethics concerns only"]

**Final Justification:**

I think the authors did a great job in addressing my concerns, and they run additional experiments that broaden the position of their work in the current landscape of T2I alignment methods.

**Limitations:**

Yes, limitations are discussed.

**Paper Formatting Concerns:**

No issues remarked.

**Quality:**

3

**Strengths And Weaknesses:**

* Strengths:

  * The authors propose a simple solution to a difficult problem, building on existing work on model alignment through direct preference optimization, through some valid insights connecting generative modeling and classification

  * The article is well written and easy to follow

  * The experimental validation goes beyond a quantitative/qualitative reporting of alignment results, and provides additional insights on the benefits of aligned models when used as classifiers, as well as an ablation study on an important hyper-parameter

* Weaknesses:

  * State-of-the-art on text-to-image alignment is assessed superficially. This works only focuses on DPO-based variants, and neglects many existing work in the domain. On the one hand, we have inference-time methods, which attempt to align T2I models by steering the generative paths through optimization of the attention layers of the score network, such as [1], and many others, on the other, we have fine-tuning methods that do not belong to the DPO family, such as [2] and [3], [4], for example.

  * The quantitative analysis of the performance of the proposed method in Table 1 relies on a protocol that hinders a proper assessment of the absolute performance of compared methods.

  * The metrics used in Table 1 are only a small subset of the many that can be used to assess alignment. See for example [2].

  * The proposed method is only compared to DPO-based alternatives, but not to other existing methods that use a different approach to alignment, such as [1-4] (and many more)

  * The proposed method, in principle, can be applied to other generative models, such as rectified flows. The evaluation is based on “older” models such as SD1.5 and SDXL, whereas the current best practice is to use FLUX or SD3, which use a different generative modeling paradigm (still related to diffusion somehow), and that are known to be more aligned than SD1.5 and SDXL.

  * There is no mention to recent work on autoregressive generative models, which appear to have superior alignment when compared to diffusion models [5, 6].

[1] Hila Chefer, Yuval Alaluf, Yael Vinker, Lior Wolf, Daniel Cohen-Or, “Attend-and-Excite: Attention-Based Semantic Guidance for Text-to-Image Diffusion Models”, https://arxiv.org/abs/2301.13826

[2] Kaiyi Huang, Chengqi Duan, Kaiyue Sun, Enze Xie, Zhenguo Li, Xihui Liu, “T2I-CompBench++: An Enhanced and Comprehensive Benchmark for Compositional Text-to-image Generation”, https://arxiv.org/abs/2307.06350

[3] Chao Wang, Giulio Franzese, Alessandro Finamore, Massimo Gallo, Pietro Michiardi, “Information Theoretic Text-to-Image Alignment”, https://arxiv.org/abs/2405.20759

[4] Ying Fan, Olivia Watkins, Yuqing Du, Hao Liu, Moonkyung Ryu, Craig Boutilier, Pieter Abbeel, Mohammad Ghavamzadeh, Kangwook Lee, Kimin Lee, “DPOK: Reinforcement Learning for Fine-tuning Text-to-Image Diffusion Models”, https://arxiv.org/abs/2305.16381

[5] Zhiyuan Yan, Junyan Ye, Weijia Li, Zilong Huang, Shenghai Yuan, Xiangyang He, Kaiqing Lin, Jun He, Conghui He, Li Yuan, “GPT-ImgEval: A Comprehensive Benchmark for Diagnosing GPT4o in Image Generation”, https://arxiv.org/abs/2504.02782v1

[6] Hu Yu, Hao Luo, Hangjie Yuan, Yu Rong, Feng Zhao, “Frequency Autoregressive Image Generation with Continuous Tokens”, https://arxiv.org/abs/2503.05305

---

> ### Author Rebuttal · Authors · 2025-07-26
>
> We greatly appreciate the reviewers' hard work and thank you for your valuable comments. We will address each  concerns in the following.
>
> ---
> **Q:  This works only focuses on DPO-based variants, and neglects many existing work in the domain. On the one hand, we have inference-time methods, which attempt to align T2I models by steering the generative paths through optimization of the attention layers of the score network**
>
> A: We believe this concern arises from a misunderstanding of the comparison scope. As discussed in *"Large Language Models Post-training: Surveying Techniques from Alignment to Reasoning,"* generative modeling methods can be broadly categorized into **pre-training** and **post-training**. Pre-training focuses on building general-purpose models from large datasets, while post-training adapts these models to specific downstream tasks.
>
> Due to their distinct goals, evaluation strategies for these two categories differ. Pre-training methods are typically compared across architecture types, datasets, or training paradigms. In contrast, post-training methods—like ours—should be evaluated using the same foundation model and training setup, with only the loss objective varying to ensure a fair comparison.
>
> Most works cited by Reviewer RW4E, except [2,3], focus on pre-training. For instance, [6] proposes an autoregressive model as a new foundation model and compares its generative capabilities with other pre-training approaches. By contrast, our ABC loss is a post-training method applied to a fixed foundation model. Analogously, comparing a fine-tuned GPT-3 with GPT-4o would not inform the quality of the fine-tuning method applied to GPT-3.
>
> Additionally, [2] is an inference-time method without training, and [3] is self-supervised, whereas DPO-based methods—including ours—require supervised training. According to our tests conducted during the rebuttal period, [2] and [3] consistently underperform all DPO-based methods on the same base model. Given their fundamentally different setups, comparing their raw performance to ours is **not meaningful** for assessing the quality of our approach.
>
> ---
> **Q: How would your approach perform with more natural prompts than the ones you used in your evaluation? Would it be possible to consider using T2I-CompBench++ for your evaluation?**
>
> A: This is a valuable suggestion. However, implementing it poses several technical challenges. T2I-CompBench++ is specifically designed to test compositional generation—handling multiple objects, attributes, and relationships—which makes it more suited for evaluating foundation models developed through **pre-training**.
>
> In contrast, **post-training** methods like ours require a well-defined downstream task and a corresponding dataset. The standard procedure involves fine-tuning a pre-trained foundation model using a specific loss and evaluating the resulting checkpoint against either the original model or other post-trained variants.
>
> In our case, we currently lack a task-specific dataset aligned with the compositional goals of T2I-CompBench++. Therefore, although T2I-CompBench++ is a promising benchmark, it is not directly applicable under our current post-training setup. Currently, we make a comporise which take the checkpoints trained on  Pick-a-Pic to conduct this evaultion. Due to the maximum length limitation, we have included the corresponding data **in our response to Reviewer qFcD.**
>
> ---
> **Q: If recent autoregressive generative models are becoming the new standard for image generation, how relevant do you consider alignment approaches for diffusion models in general, and DPO-based fine-tuning approaches in particular, will be?**
>
> A: Both autoregressive generative models and diffusion models are promising approaches for image generation. The DPO (Differentiable Prompt Optimization) method was initially proposed for aligning autoregressive language models, and Diffusion-DPO adapts this technique for use with diffusion models. The primary contribution of our paper is that we are the first to demonstrate that the alignment task can be transformed into a classification task. Reviewer RW4E can verify that this discovery does not depend on the model type. Both autoregressive and diffusion models can benefit from this finding.
>
> Since Reviewer RW4E has a particular interest in the application of the ABC loss to autoregressive generative models, we believe it would be a great idea to prepare a separate paper discussing how to apply the ABC loss to autoregressive models.
>
> ---
> **Q: What are the key advantages of DPO-based alignment methods such as ABC when compared to other fine-tuning methods from the literature (e.g., you cite DPOK as reference [14], but do not even refer to it in the main), or other inference-time methods?**
>
> A: Both DPO and SFT methods have demonstrated their value in training large language models. SFT-based methods, such as DPOK, require a reward model to fine-tune the diffusion model. However, training a good reward model requires additional time and dataset resources. In contrast, DPO-based methods bypass the need for a reward model when fine-tuning the diffusion model.
>
> Regarding inference-time methods, the running time for inference is significantly longer compared to SFT and DPO-based methods. In general, these three approaches represent different pathways toward achieving better results, and at this stage, it is difficult to predict which will ultimately prevail. This paper focuses solely on demonstrating that the alignment task can be transformed into a classification task through our ABC loss. We do not address whether DPO-based methods will outperform SFT and inference-based methods, as this is beyond the scope of our current work.
>
> ---
>
> **Q: In sec 4.1, line 116, you rely on a pretty strong assumption. I realize your work is not the only one requiring such an assumption, but it would be interesting to hear your opinion on how a DPO-based framework such as ABC could cope with the more realistic case of multiple images being aligned to a single prompt.**
>
> A: Thank you for your question. We acknowledge the limitation of this assumption. However, we could not avoid it in the current environment for two reasons:
>
> 1. The current preference datasets are built on this assumption.
> 2. The evaluation networks are also designed based on this assumption, as they are trained on the preference dataset.
>
> Without new preference datasets and evaluation criteria, it is difficult for methods that handle multiple preferences to demonstrate their superiority and gain acceptance in top conferences like NeurIPS.
>
> Since we have established the connection between alignment and classification, the assumption that "each text prompt corresponds to a single aligned image" can be interpreted as a multi-class classification task, where each sample is assumed to belong to only one class. In contrast, Reviewer RW4E's preference for a scenario where one prompt corresponds to multiple images can be interpreted as a multi-label classification task, where each sample may belong to multiple classes. In the future, we plan to explore the multi-label classification perspective to address this issue.
>
> ---
>
> **Q: In Theorem 2, lines 142-143, this assumption is pretty strong. Can you comment on what would it be necessary to do if we relax such an assumption?**
>
> A: We apologize for any confusion caused to Reviewer RW4E. In our paper, Theorem 2 is used to illustrate that classification performance directly determines alignment performance. To demonstrate this, we assume that we have a total of $N$ text prompts $\mathrm{y}_i$​, each corresponding to an aligned image $\pmb{x}\_{\mathrm{y}\_i} $​​. We believe it is reasonable to assume that the prior distribution for both the prompt $\mathrm{y}_i$​ and the image $\pmb{x}\_{\mathrm{y}\_i} $​​ follows a uniform distribution. If we were to assume otherwise, we would have to acknowledge that certain labels and images are inherently more special than others. However, we currently do not have a compelling reason to support this assumption.
>
> ---
> **Q: The proposed method, in principle, can be applied to other generative models, such as rectified flows. The evaluation is based on “older” models such as SD1.5 and SDXL, whereas the current best practice is to use FLUX or SD3, which use a different generative modeling paradigm (still related to diffusion somehow), and that are known to be more aligned than SD1.5 and SDXL.**
>
> A: Applying DPO-like methods to flow models is not an easy task. Flow models rely on a deterministic generative process based on Ordinary Differential Equations (ODEs), meaning they cannot sample stochastically during inference. In contrast, DPO relies on stochastic sampling to explore the environment, learning by trying different actions and improving based on rewards. This need for stochasticity conflicts with the deterministic nature of flow matching models. To apply DPO to flow models, one would need to conduct an ODE-to-SDE conversion, which transforms a deterministic ODE into an equivalent Stochastic Differential Equation (SDE) that matches the original model’s marginal distribution at all timesteps, thus enabling statistical sampling.
>
> We recently discovered a concurrent work, "Flow-GRPO: Training Flow Matching Models via Online RL", which claims to be the first method to apply GRPO to flow models in order to address this problem. By employing  Flow-GRPO, our approach could potentially be applied to models like FLUX or SD3. However, this paper was released on May 8, 2025, on arXiv, and the code was released on May 15, 2025. As a result, we were not aware of this work before submitting our paper.
>
> ---
> **Q: Could you report absolute metrics rather than percentage of wins in Table 1?**
>
> A: Thank you for the suggestion. Due to length limits, we report the absolute metrics in Table 2 of our response to Reviewer PucN, including appended results for DPOK and D3PO.

---

> > ### Comment · Reviewer_RW4E · 2025-08-01
> > **Thank you for the rebuttal**
> >
> > Dear authors,
> > thank you very much for your rebuttal, and hard work in producing additional results (I have read other reviews and your response).
> >
> > While I appreciate the technical contribution of this work, I do not agree on the narrow angle taken by the authors concerning the empirical evaluation and comparison to alternative methodologies to address the T2I alignment problem. In my opinion the evaluation is limited in 1) comparison to other methods (inference time, other fine-tuning approaches such as MITUNE), 2) prompt complexity (e.g. using natural prompts). Although the authors indicate superior performance to alternative such as [2] and [3], at the time of this writing, I do not have enough information to properly assess such claim.
> >
> > I will remain open during the discussion phase, both with authors and other reviewers, to revise my score and to avoid "blocking" an interesting idea to be published and discussed in the community. However, as of now, I will not change my score.

---

> ### Author Response · Authors · 2025-08-04
>
> Dear Reviewer RW4E,
>
> We sincerely apologize for the delayed response, as we needed additional time to carefully consider how to conduct a fair comparison. We also acknowledge that our initial rebuttal did not provide sufficient evidence to fully address your concerns. We would like to take this opportunity to clarify why the comparisons you suggested were not included. Our decision was based on three main reasons: (1) the referenced methods belong to different methodological categories; (2) cross-category comparisons raise concerns regarding fairness and interpretability; and (3) space limitations prevented us from including additional results.
>
> For the first reason, we note that A&E [1] is a training-free approach designed to improve inference-time performance without modifying model weights. In contrast, methods [3], [4], and ours are fine-tuning-based and rely on data-driven checkpoint updates. Even among fine-tuning methods, the assumptions differ significantly: MITUNE [3] generates pseudo-labeled data automatically using the base model, while method [4] assumes access to human-annotated data and sufficient resources to train a reward model, which is then used to guide the training of the generation model. In resource-constrained environments—especially those with limited GPU memory—method [4] becomes impractical, making DPO-style methods like ours the only viable option. Furthermore, the training data used also differs: DPO is trained on preference triplets (prompt, preferred image, less preferred image), whereas \[4] uses prompt-image pairs without explicit preference signals. These differences in both input requirements and training objectives further complicate direct comparison.
>
> For the second reason, even when technically possible, fair comparisons remain challenging. Fine-tuning methods yield new checkpoints that are evaluated under a shared sampling strategy, while training-free methods like A&E [1] modify the sampling process itself, operating on a fixed checkpoint. As such, the natural baselines for A&E [1] are other sampling strategies (e.g., DDPM, DDIM), not fine-tuned models.
>
> To aid your evaluation, we report the performance of our fine-tuned checkpoint under both DDPM and A&E \[1]’s sampling strategy, along with the performance of the original (unmodified) SDXL checkpoint under the same conditions. Note that A\&E \[1] requires manually specified subject tokens to guide inference—information that standard DDPM cannot leverage—further limiting the validity of direct comparisons.
>
> |Model|Color (B-VQA)|Shape (B-VQA)|Texture (B-VQA)|Numeracy (UniDet)|2D-Spatial (UniDet)|3D-Spatial (UniDet)|Non-Spatial (CLIP)|
> |-|-|-|-|-|-|-|-|
> |SDXL+DDPM|0.5708|0.4880|0.5600|0.5591|0.1949|0.3551|0.3065|
> |SDXL+A&E|0.6589|0.5101|0.6737|0.5319|0.2272|0.3602|0.3204|
> |ABC+DDPM|0.6708|0.5450|0.6866|0.5623|0.2401|0.3697|0.3154|
> |ABC+A&E|0.6936|0.5726|0.7023|0.5744|0.2368|0.3745|0.3256|
>
> We also compare our fine-tuned model (ABC) against MITUNE \[3] under the same DDPM sampling protocol:
>
> |Model|PickScore (P)|HPS (P)|Aesth. (P)|CLIP (P)|PickScore (H)|HPS (H)| Aesth. (H)|CLIP (H)|
> |-|-|-|-|-|-|-|-|-|
> |SDXL- MITUNE|21.61±2.22|27.04±3.26|5.54±0.74|30.11±8.56|23.06±2.47|29.80±2.98|5.92±0.99|39.96±5.90|
> |SDXL-ABC|23.79±2.27|29.42±3.29 |6.35±0.89|36.81±8.51|24.39±2.38|30.67±3.08|6.54±0.86|38.97±6.02|
>
> We further provide a comparison on **T2I-CompBench++**, showing that ABC consistently outperforms MITUNE across most dimensions:
>
> |Model|Color (B-VQA)|Shape (B-VQA)|Texture (B-VQA)|Numeracy (UniDet)|2D-Spatial (UniDet)|3D-Spatial (UniDet)|Non-Spatial (CLIP)|
> |-|-|-|-|-|-|-|-|
> |MITUNE+DDPM|0.6912|0.5261|0.6608|0.4866|0.2353|0.3472|0.3174|
> |ABC+DDPM |0.6708|0.5450|0.6866|0.5623|0.2401|0.3697|0.3154|
>
> Overall, our method consistently outperforms MITUNE across a wide range of metrics. We attribute this to two main factors: (1) our dataset includes human-annotated preferences, whereas MITUNE relies on synthetic labels generated by the base model; and (2) our proposed ABC loss is more effective than the pseudo-labeling strategy employed by MITUNE.
>
> We acknowledge that on T2I-CompBench++, ABC underperforms MITUNE on two specific scores. We attribute this to differences in the training datasets: the dataset used for ABC was not specifically designed for the T2I-CompBench++ task, whereas the dataset used in MITUNE was optimized to maximize point-wise mutual information—that is, to emphasize the difference between conditional and unconditional matching scores. While this design benefits pixel-level fidelity, it may lead to weaker semantic representation and alignment, which in turn affects performance on certain benchmarks.
>
> Please feel free to reach out if you have any further questions. We sincerely appreciate your thoughtful feedback.
>
> Best regards,
>
> Authors

---

> > ### Comment · Reviewer_RW4E · 2025-08-06
> > **Thank you for the additional experiments**
> >
> > Dear authors,
> > thank you for the additional experiments and discussion. I think this definitely strengthen your paper, and provides a broader view on the approach you proposed.
> > I will raise my score.
> >
> > Thanks

---

> > > ### Author Response · Authors · 2025-08-07
> > >
> > > Dear Reviewer RW4E
> > >
> > > We sincerely thank you for the positive feedback and for recognizing the value of the additional experiments and discussions. We are pleased to hear that these additions have helped strengthen the paper and provide a broader perspective on our proposed approach. We greatly appreciate your support and are grateful for the updated score.
> > >
> > > Best Wishes!
> > >
> > > The Authors

---

### Official Review · Reviewer_PucN · 2025-07-01

**Clarity:** 3
**Significance:** 3
**Originality:** 3
**Rating:** 5
**Confidence:** 4

**Summary:**

This work proposes a novel connection between preference alignment and classification, i.e. being able to discriminate between preferred and unpreferred samples, and presents an alignment objective for aligning text-to-image diffusion models. The objective extends the Circle Loss objective into the setting of text-to-image diffusion models where the score, provided by the diffusion model, is used to approximate p(y|x). Unlike related approaches like Diffusion-DPO (which ABC, their method, is connected to), the provided objective does not require a reference model during training, and thus provides some memory/compute benefits as well. Results show that ABC outperforms prior alignment approaches according to existing reward models and human annotators.

**Questions:**

- Can the authors report the statistical significance of the quantitative preference alignment experiments?

- For the quantitative preference experiment using off-the-shelf reward models, can the authors report the average score (within some confidence interval) to help quantify the improvement of ABC over other methods?

- Can the authors provide a quantitative ablation studying the effect of the data augmentation strategy?

**Ethical Concerns:**

["NO or VERY MINOR ethics concerns only"]

**Final Justification:**

The paper presents an interesting alternative to T2I alignment by formulating it is a classification problem. The results provided in the rebuttal are statistically significant (spec. for SDXL) and show the benefit of this approach. While the loss function is not novel, I believe it's application to this new domain and its performance merits acceptance.

**Limitations:**

yes

**Quality:**

3

**Strengths And Weaknesses:**

**Strengths**

- Proposes novel connection between classification and alignment, that is justified both theoretically and empirically.

- Does not require a reference model during training, saving memory & compute

- Strong, comprehensive results on automated evaluations (e.g. using HPS reward model as judge) and via user study.

**Weaknesses**

- Importance of data augmentation is unclear. The authors note that the training loss may diverge without the use of data augmentation. However, the effect of this augmentation is not well documented, e.g. there is no quantitative ablation on the effect of the augmentation.

- Strong assumptions regarding the nature of preference data, e.g. each text prompt corresponds to one aligned image, or the data augmentation strategy that prefaces the losing image’s prompt with “The image that aligns less with human preferences”. Real-world preference data is quite noisy and its unclear why a formulation dependent on such strong assumptions is able to outperform other methods like Diffusion-KTO, for instance, which have some noise-resistant properties. Additionally, the data augmentation strategy is questionable as even though an image is less-preferred in a single triplet it may generally be more preferred on aggregate. It is unclear why this strategy works with noisy real-world data.

- Statistical significance of results. The authors report win-rates in the quantitative preference alignment experiments. However, win-rate can be quite sensitive to the choice of seed, noise, etc, as in order for a method to win it just needs to at least very slightly outperform the other per the reward model score. To fully validate the performance of ABC, statistical significance of these results should be provided, e.g. via confidence intervals.

- Lack of user study details. The details of the user study are largely omitted in the paper, outside of 2 lines in the checklist. Was this a blind user study? How many annotators were assigned per comparison?

- [minor editorial] Line 75 is missing a citation for triplet loss.

---

> ### Author Rebuttal · Authors · 2025-07-26
>
> We greatly appreciate the reviewers' hard work and thank you for your valuable comments. We will address each  concerns in the following.
>
> ---
> **Q: Importance of data augmentation is unclear.**
>
> A: We apologize for the confusion regarding data augmentation in the preference data. Each dataset item is a triplet $(\mathrm{y}, \pmb{x}^+\_{\mathrm{y}}, \pmb{x}^-\_{\mathrm{y}})$, where $\mathrm{y}$ denotes the prompt, and $\pmb{x}^+\_{\mathrm{y}}$ and $\pmb{x}^-\_{\mathrm{y}}$ represent the positive and negative images corresponding to the prompt $\mathrm{y}$, respectively.
>
> In this paper, we propose replacing the alignment loss (DPO loss) with a classification loss (ABC loss). Since the classification loss requires a label for each image, we need to transform the triplet $(\mathrm{y}, \pmb{x}^+\_{\mathrm{y}}, \pmb{x}^-\_{\mathrm{y}})$, where the two images share the same label $\mathrm{y}$, into a form where each image has its own label. Specifically, we define $\mathrm{y}^+$ as the original prompt $\mathrm{y}$ and construct $\mathrm{y}^-$ by appending “The image that aligns less with human preferences” to $\mathrm{y}$. This reformulation transforms each preference tuple $(\mathrm{y}, \pmb{x}^+_{\mathrm{y}}, \pmb{x}^-\_{\mathrm{y}})$ into two supervised examples: $(\mathrm{y}^+, \pmb{x}^+\_{\mathrm{y}})$ and $(\mathrm{y}^-, \pmb{x}^-\_{\mathrm{y}})$. This is clarified in lines 173 to 176 of the paper.
>
> ---
> **Q: Real-world preference data is quite noisy and its unclear why a formulation dependent on such strong assumptions is able to outperform other methods like Diffusion-KTO. Additionally, the data augmentation strategy is questionable as even though an image is less-preferred in a single triplet it may generally be more preferred on aggregate. It is unclear why this strategy works with noisy real-world data.**
>
> A: Yes, this is a great question. Preference data is inherently noisy, making it difficult to establish a clear principle for determining whether one image is preferred over another. Current diffusion alignment methods do not take this issue into account. To address this problem, we believe it is necessary to build a new dataset that provides preference comparison results across different dimensions. However, current alignment methods are typically trained on datasets like Pick-a-Pic, which assumes that one image is better than another with respect to the given prompt.
>
> We do not directly address this issue in the current paper, but we present a promising approach to tackle it. Specifically, we show that the alignment task can be transformed into a classification problem, and classification in noisy environments has been extensively studied. It may therefore be possible to adapt existing methods from noisy classification to address the noisy alignment problem.
>
> As for the noise-resistant properties of our method, they are easy to interpret. The well-known ImageNet classification dataset contains incorrect labels, yet research has shown that simply using cross-entropy loss to train the network still yields state-of-the-art performance, without the need for special handling of mislabeled data. We attribute this noise-resistant property to the classification loss. In this paper, we transform the preference data into a classification form. As a result, even if the data is noisy, the network retains the ability to resist the effects of noise.
>
> ---
> **Q: Statistical significance of results. To fully validate the performance of ABC, statistical significance of these results should be provided, e.g. via confidence intervals.**
>
> A: Thank you for raising this important point. We agree that providing statistical significance is important for validating the reported win rates. While our experiments followed the standard protocols used in prior work, we now include win rates along with variance intervals.
>
> Specifically, we compute these intervals by discarding the top 5% of deviations from the mean, resulting in a range that captures 95% of the scores. This provides a robust estimate of variability and offers a more informative view of our method’s performance.
>
> **Table 1.** Win rate  on PartiPrompts (P) and HPS (H) benchmarks for SD1.5 and SDXL-based models.
>
> | Model      | PickScore (P) | HPS (P)       | Aesth. (P)    | CLIP (P)      | PickScore (H) | HPS (H)       | Aesth. (H)    | CLIP (H)      |
> | ---------- | ------------- | ------------- | ------------- | ------------- | ------------- | ------------- | ------------- | ------------- |
> | SD1.5-Base | 60.02 ± 2.40% | 81.51 ± 2.21% | 74.27 ± 2.14% | 59.72 ± 2.45% | 74.83 ± 3.10% | 85.75 ± 2.87% | 68.84 ± 3.31% | 59.65 ± 3.53% |
> | SD1.5-DPOK  |57.15 ± 2.45% | 68.19 ± 2.21% | 62.51 ± 2.42% | 55.28 ± 2.45% | 52.17 ± 3.53% | 68.16 ± 3.46% | 65.24 ± 3.49% | 60.95 ± 3.56% |
> | SD1.5-D3PO  | 58.42 ± 2.43% | 72.75 ± 2.30% | 65.72 ± 2.40% | 53.61 ± 2.45% | 52.33± 3.52% | 73.25 ± 3.46% | 63.89 ± 3.51% | 54.14 ± 3.62%|
> | SD1.5-DPO  | 55.85 ± 2.44% | 73.02 ± 2.38% | 64.90 ± 2.34% | 44.97 ± 2.43% | 53.46 ± 3.51% | 71.50 ± 3.46% | 64.19 ± 3.56% | 52.06 ± 3.57% |
> | SD1.5-SPO  | 51.16 ± 2.41% | 61.59 ± 2.41% | 47.60 ± 2.45% | 60.02 ± 2.36% | 45.35 ± 3.57% | 54.99 ± 3.53% | 38.08 ± 3.47% | 64.83 ± 3.13% |
> | SD1.5-KTO  | 57.77 ± 2.41% | 44.72 ± 2.38% | 53.90 ± 2.44% | 47.22 ± 2.45% | 52.28 ± 3.52% | 42.88 ± 3.48% | 52.86 ± 3.57% | 53.93 ± 3.55% |
> |            |               |               |               |               |               |               |               |               |
> | SDXL-Base  | 74.38 ± 1.84% | 79.26 ± 1.97% | 80.20 ± 1.56% | 52.46 ± 2.04% | 79.35 ± 2.23% | 70.17 ± 2.58% | 72.28 ± 2.86% | 60.38 ± 3.26% |
> | SDXL-DPO   | 73.22 ± 2.18% | 72.50 ± 2.33% | 68.25 ± 1.39% | 50.51 ± 1.88% | 77.26 ± 3.06% | 69.54 ± 3.42% | 70.19 ± 2.39% | 57.06 ± 3.07% |
> | SDXL-SPO   | 52.49 ± 2.34% | 40.31 ± 2.44% | 59.93 ± 2.41% | 55.53 ± 2.44% | 51.16 ± 3.34% | 52.41 ± 3.57% | 46.78 ± 3.53% | 59.87 ± 3.58% |
> | SDXL-MAPO  | 65.35 ± 1.81% | 81.17 ± 2.07% | 72.10 ± 1.82% | 46.97 ± 2.06% | 68.55 ± 2.37% | 64.89 ± 3.11% | 68.18 ± 2.96% | 51.14 ± 3.19% |
>
>
> ---
> **Q: Lack of user study details. The details of the user study are largely omitted in the paper, outside of 2 lines in the checklist. Was this a blind user study? How many annotators were assigned per comparison?**
>
> A: Thank you for your question. We conducted a user study to compare the proposed ABC method with several baseline approaches. Specifically, we randomly sampled 100 prompts from the PartiPrompts dataset and another 100 prompts from the HPSv2 benchmark. For each prompt, we generated five images using five different methods.
>
> Participants were shown five images per prompt (one from each method) and asked to answer three questions:
>
> 1. Which image is your overall preferred choice?
> 2. Which image is more visually attractive?
> 3. Which image better matches the text description?
>
> To minimize position bias, the order of images was randomized for each prompt. Each method’s final score was computed as a weighted sum of its win rates under the three criteria, with weights of 30% for general preference, 30% for visual appeal, and 40% for prompt alignment.
>
> The study was conducted as a blind evaluation. Annotators were not informed about which method generated each image. We recruited participants from our research group, comprising approximately 100 students, and collected a total of 82 valid responses.
>
> We hope this clarification provides a more complete picture of our user study design and evaluation protocol.
>
> ---
> **Q: Can the authors report the average score (within some confidence interval) to help quantify the improvement of ABC over other methods?**
>
> A: We thank the reviewer for the valuable suggestion. In addition to win-rate comparisons, we now report the absolute scores along with the setting in the Table 1.
>
> **Table 2.** Absolute scores  on the PartiPrompts (P) and HPS (H) benchmarks for SD1.5 and SDXL-based models.
>
> |Model|PickScore (P)|HPS (P)| Aesth. (P)|CLIP (P)|PickScore (H)|HPS (H)|Aesth. (H)|CLIP (H)|
> |-|-|-|-|-|-|-|-|-|
> |SD1.5-Base|21.25±2.02|26.98±2.85|5.29±1.17|29.57±9.71|21.17±2.45|27.61±3.10|5.45±1.01|35.73±7.38|
> |SD1.5-DPOK|21.57±2.21|27.21±3.03|5.59±1.21|30.01±9.93|21.80±2.42|28.15±3.17|5.60±1.04|36.75±8.12|
> |SD1.5-D3PO|21.41±2.03|27.08±2.97|5.42±1.13|29.87±8.66|21.76±2.58|28.20±3.23|5.54±1.07|36.71±7.87|
> |SD1.5-DPO|21.49±2.19|27.16±3.13|5.36±1.08|29.80±9.46|21.71±2.41|28.23±3.35|5.59±0.99|36.66±7.95|
> |SD1.5-SPO|21.53±2.33|27.33±3.74|5.89±1.04|28.13±9.23|21.99±2.76|28.53±3.69|5.96±1.10|33.14±9.15|
> |SD1.5-KTO|21.46±1.96|27.70±3.21|5.62±0.99|30.78±8.09|21.79±2.53|28.95±3.12|5.62±0.96|37.01±8.01|
> |SD1.5-ABC|21.85±2.04|27.97±2.85|5.93±1.01|31.07±8.14|21.97±2.45|28.86±3.02|5.72±0.87|37.18±7.72|
> ||||||||||
> |SDXL-Base|22.76±2.45|28.49±3.59|5.86±1.05|35.76±9.66|23.26±2.50|29.38±3.59|6.08±1.03|37.24±6.74|
> |SDXL-DPO|22.94±2.37|28.93±3.52|6.01±0.98|36.01±8.73|23.59±2.65|29.86±3.40|6.14±1.00|38.34±5.98|
> |SDXL-SPO|23.56±2.64|29.12±3.52|6.26±0.92|33.82±9.84|23.76±2.70|30.30±3.18|6.48±0.86|37.62±7.14|
> |SDXL-MAPO|22.82±2.40|28.62±3.62|5.98±1.04|36.58±9.39|23.60±2.53|29.92±3.52|6.19±0.92|38.61±7.17|
> |SDXL-ABC|23.79±2.27|29.42±3.29|6.35±0.89|36.81±8.51|24.39±2.38|30.67±3.08|6.54±0.86|38.97±6.02|

---

> > ### Comment · Reviewer_PucN · 2025-08-05
> >
> > I would like to thank the authors for their work and for answering all of my questions.

---

### Official Review · Reviewer_qFcD · 2025-07-03

**Clarity:** 2
**Significance:** 3
**Originality:** 3
**Rating:** 4
**Confidence:** 3

**Summary:**

This paper proposes a novel loss function for aligning text-to-image diffusion models with human preferences. The authors hypothesize that conventional DPO-type loss functions are limited due to their reliance on a reference model. Since the reference model itself is not aligned with human preferences, comparisons against it may lead to sub-optimal solutions. To address this, the authors introduce a new loss function called ABC (Alignment by Classification) loss, which does not require a reference model. ABC loss resembles a contrastive loss that directly compares positive and negative pairs. The authors first reformulate DPO with an ideal reference model as a classification problem within the diffusion framework and argue that alignment performance depends on the model’s discriminative capability. Based on this insight, they design the ABC loss. Experimental results demonstrate that the proposed approach achieves better alignment with human preferences compared to existing loss functions.

**Questions:**

please check the weaknesses

**Ethical Concerns:**

["NO or VERY MINOR ethics concerns only"]

**Final Justification:**

The authors have adequately addressed my concerns, and I will maintain my positive score. However, my confidence in the evaluation remains low.

**Limitations:**

Yes

**Quality:**

3

**Strengths And Weaknesses:**

Pros

1. The paper proposes a novel loss function for aligning text-to-image diffusion models with human preferences, and it outperforms conventional loss functions.

2. The semi-supervised learning technique introduced in Section 4.2 is interesting and appears effective.

3. The authors provide various qualitative examples and compare their method with recent approaches.

Cons

1. The paper is not easy to read, particularly the theorem sections. It would be helpful if the authors could provide informal interpretations or explanations in simpler terms to improve accessibility.

2. Due to the large number of notations, the paper is difficult to follow. Including a table summarizing all notations and their meanings would greatly enhance readability.

3. Additionally, comparisons of training time and analyses of the effect of dataset size are important. More extensive ablation studies exploring different aspects could further strengthen the paper.

---

> ### Author Rebuttal · Authors · 2025-07-26
>
> We greatly appreciate the reviewers' hard work and thank you for your valuable comments. We will address each  concerns in the following.
>
> ---
> **Q: The paper is not easy to read, particularly the theorem sections. It would be helpful if the authors could provide informal interpretations or explanations in simpler terms to improve accessibility.**
>
> A: We apologize that the theorem sections may have made the paper difficult to follow. We briefly interpret them here. **Theorem 1** proves that the AM-Softmax loss is upper bounded by the Diffusion-DPO loss. In other words, minimizing the Diffusion-DPO loss for better alignment will also reduce the AM-Softmax loss, leading to improved classification performance. Simply put, **better alignment leads to better classification**. Conversely, **Theorem 2** shows that, under certain conditions, improved classification leads to better alignment. Together, these two theorems reveal a strong connection between classification and alignment, forming the theoretical foundation of our approach, which replaces the DPO loss with the ABC loss for alignment tasks.  We first establish a connection between alignment and classification. This idea is conceptually similar to the ICLR 2025 paper "On a Connection Between Imitation Learning and RLHF," which draws a connection between Reinforcement Learning from Human Feedback (RLHF) and Imitation Learning.
>
> ---
> **Q: Due to the large number of notations, the paper is difficult to follow. Including a table summarizing all notations and their meanings would greatly enhance readability.**
>
> A: We apologize for the oversight in summarizing the notations. Below is a table that summarizes all the notations used in the paper:
>
> | Symbol        | Description   |
> | :--------  | :-----  |
> | $x_0$ |Clean image sampled from real data distribution $q(x_0)$. |
> | $x_y^±$ |Clean image $x_0$ aligned ($+$) or misaligned ($-$) with prompt $y$ in preference pairs. |
> | $x^±_{t;y}$ |Noisy version of  $x_y^±$ at timestep $t$. |
> | $x_{y^±}$ |The same image $x$ conditioned on aligned ($+$) and misaligned ($-$) prompts $y^±$. |
> | $\epsilon$|Ground-truth noise sampled from $\mathcal{N}(0, \mathbf{I})$, added during forward diffusion. |
> | $\epsilon_\theta(x,t)$|Noise predicted by the model $\epsilon_\theta$ at timestep $t$, given noisy input $x$. |
> | $\epsilon_\mathrm{ali}(x,y,t)$ | Noise prediction from the ideal alignment model for input image $x$, prompt $y$ and timestep $t$. |
> | $\epsilon_\mathrm{ref}(x,y,t)$ |Noise predicted by the reference diffusion model for input image $x$, prompt $y$ and timestep $t$. |
> | $\epsilon_\mathrm{opt}(x,y,t)$ |Optimal noise prediction as a weighted average over clean images, obtained via marginalization of $p(x \mid y)$. |
> | $s_{\theta}(x,y)$ | Score predicted by the traing model for image sample $x$ and prompt $y$ |
> | $s_\mathrm{ref}(x,y)$ |Score predicted by the reference model for image sample $x$ and prompt $y$  |
> | $\delta$ | Margin used to enforce separation between aligned ($+$) and misaligned ($-$) scores. |
> | $\Delta^±_{y}$ |Margin offsets applied to the expected scores of aligned ($+$) and misaligned ($-$) images with prompt ${y}$. |
> |$O^±_y$|Ideal score targets for aligned ($+$) and misaligned ($-$) images with prompt ${y}$.|
> | $\eta^±_{y}$ | Scaling factors that modulate the contributions of aligned ($+$) and misaligned ($-$) images with prompt ${y}$. |
> | $\iota({x}^-_y, {x}^+_y, {y})$ | Score difference in ABC loss between aligned ($+$) and misaligned ($-$) images with prompt ${y}$. |
>
>
> PS：
>
> 1. $x_0$ can refer to any clean image, while $x_y^+$, $x_y^-$ refer specifically to images aligned/misaligned with a prompt $y$.
> 2. For simplicity, the timestep $t$ is sometimes omitted in notation when it is clear from context, eg. $\epsilon_\mathrm{ali}(x,y)$.
> 3. For simplicity, the prompt $y$ is sometimes omitted in notation , eg. $\iota({x}^-, {x}^+, {y})$.
>
> ---
>
> **Q:  Comparisons of training time and analyses of the effect of dataset size are important. More extensive ablation studies exploring different aspects could further strengthen the paper.**
>
> A: Thank you for the suggestion. We believe it will significantly improve our paper. Due to the page limitations of NeurIPS, we initially restricted the training dataset to Pick-a-Pic v2, published in "Pick-a-Pic: An Open Dataset of User Preferences for Text-to-Image Generation," which is commonly adopted by current diffusion alignment methods. Additionally, given the current research focus on designing various alignment losses, the training overhead for these methods is minimal. As a result, the training time for different alignment methods remains nearly the same for any given stable diffusion model. This helps explain why we did not include an ablation study comparing training times or analyzing the effect of dataset size.
>
> ----
>
> **We sincerely apologize for the inconvenience.  Following is the supplementary response for Reviewer RW4E, provided here due to the rebuttal length limitation.**
>
> ----
>
> Q: How would your approach perform with more natural prompts than the ones you used in your evaluation? Would it be possible to consider using T2I-CompBench++ for your evaluation?
>
> A: Following the official setup, we tested on 8,000 prompts in total, covering: attribute binding, relationships, numeracy, and complex compositions. We report the results in the tables below. Higher scores indicate better alignment with the intended composition.
>
> | Model      | Color (B-VQA) |Shape (B-VQA) |Texture (B-VQA) | Numeracy (UniDet) | 2D-Spatial (UniDet) | 3D-Spatial (UniDet) | Non-Spatial (CLIP) | Complex (3-in-1) |
> | ---------- | ---------- | ---------- | ------------ | -------- | ---------- | ---------- | ----------- | ------- |
> | SD1.5-Base | 0.3811     | 0.3395     | 0.4192       | 0.4436   | 0.1460     | 0.2912     | 0.3092      | 0.3002  |
> | SD1.5-DPO  | 0.3943     | 0.3440     | 0.4374       | 0.4523   | 0.1627     | 0.3090     | 0.3091      | 0.3032  |
> | SD1.5-SPO  | 0.4030     | 0.4001     | 0.4152       | 0.4461   | 0.1471     | 0.2958     | 0.3010      | 0.3131  |
> | SD1.5-KTO  | 0.4645     | 0.3815     | 0.4730       | 0.4618   | 0.1919     | 0.3318     | 0.3104      | 0.3514  |
> | SD1.5-ABC  | 0.4647     | 0.4005     | 0.4751       | 0.4570   | 0.1895     | 0.3324     | 0.3106      | 0.3587  |
> |  |  |  |  |  |  |  |  |  |
> | SDXL-Base | 0.5708     | 0.4880     | 0.5600       | 0.5591   | 0.1949     | 0.3551     | 0.3065      | 0.4383  |
> | SDXL-DPO  | 0.6586     | 0.5358     | 0.6521       | 0.5300   | 0.2376     | 0.3668     | 0.3116      | 0.4923  |
> | SDXL-SPO  | 0.6431     | 0.5200     | 0.6496       | 0.5765   | 0.2298     | 0.3513     | 0.3031      | 0.4424  |
> | SDXL-MAPO | 0.6682     | 0.5104     | 0.5650       | 0.5189   | 0.1700     | 0.3507     | 0.3136      | 0.4401  |
> | SDXL-ABC  | 0.6708     | 0.5450     | 0.6866       | 0.5623   | 0.2401     | 0.3697     | 0.3154      | 0.5051  |

---

> > ### Comment · Reviewer_qFcD · 2025-08-02
> >
> > The authors have addressed my concerns, so I will maintain my score. However, I am unsure if it is entirely appropriate for the authors to use their rebuttal to address another reviewer’s comments in response to mine.

---

> > > ### Author Response · Authors · 2025-08-04
> > >
> > > Thank you for your feedback and for maintaining your score.

---

### Note · Authors · 2025-08-14

**Dear Area Chair and Reviewers,**

We sincerely thank you again for your time and effort in reviewing our work. Our paper initially received scores of 4, 4, and 3, and Reviewer RW4E has kindly confirmed raising their score from 3. During the rebuttal period, we carefully addressed all concerns and received positive feedback from all reviewers.

Below, we outline the planned revisions to further strengthen the paper:

* We additionally evaluated our method on **T2I-CompBench++** (8,000 prompts) and achieved consistently higher scores than DPO-based baselines, demonstrating that the improvements generalize beyond the original benchmarks.
* We compared our method with non-DPO-based approaches, including inference-time methods (Attend-and-Excite) and fine-tuning methods (MITUNE) under matched evaluation protocols.
* We will add a **table of notations** in the appendix and provide more accessible explanations of the two theorems to improve clarity and readability.
* We will expand the description of the **user study** in the “Quantitative Comparison” section to clearly present its design and evaluation protocol.
* We will clarify the role of the **data augmentation step** in reformulating preference tuples for the classification loss and explain why it is necessary in our setting.
* We will add **confidence intervals** for the reported win rates in Table 1 to indicate statistical significance and include an additional table reporting absolute scores (with confidence intervals) to complement the win-rate comparisons.
* We will correct all typos and formatting issues and add the missing citation for **triplet loss** in Line 75.

We are deeply grateful for your constructive feedback, which has significantly improved the quality of our work. Thank you again for your time, careful review, and valuable suggestions.

Best regards,

**The Authors**

---

### Decision · Program_Chairs · 2025-09-17

**Decision:**

Accept (spotlight)

**Comment:**

This paper proposes a genuinely new approach to the problem of T2i diffusion model alignment (an important problem with wide applications). It recasts the problem as a certain kind of classification problem, using ideas from circle loss. All reviewers appreciated the novelty and interestingness of this idea. A significant advantage of the method is that a reference model is NOT needed during training which saves a lot of memory, a very significant factor for frontier diffusion models. Objections mainly focused on the robustness of the evaluations, many of which were effectively addressed by the authors with new experiments etc. Overall, a solid work that is likely to influence new ideas and directions.